# Clinical application of HEDI for biomechanical evaluation and visualisation in incisional hernia repair
Philipp D. Lösel [1,12] ✉, Jacob J. Relle [2,3,12], Samuel Voß [4], Ramesch Raschidi[5], Regine Nessel[6], Johannes Görich [7], Mark O. Wielpütz [8,9], Thorsten Löffler[10], Vincent Heuveline [2,3] & Friedrich Kallinowski [11]

## Abstract

**Background** Abdominal wall defects, such as incisional hernias, are a common source of pain and discomfort and often require repeated surgical interventions. Traditional mesh repair techniques typically rely on fixed overlap based on defect size, without considering important biomechanical factors like muscle activity, internal pressure, and tissue elasticity. This study aims to introduce a biomechanical approach to incisional hernia repair that accounts for abdominal wall instability and to evaluate a visualisation tool designed to support surgical planning.

**Methods** We developed HEDI, a tool that uses computed tomography with Valsalva manoeuvre to automatically assess hernia size, volume, and abdominal wall instability. This tool was applied in the preoperative evaluation of 31 patients undergoing incisional hernia repair. Surgeries were performed concurrently with the development of the tool, and patient outcomes were monitored over a three-year period.

**Results** Here we show that all 31 patients remain free of pain and hernia recurrence three years after surgery. The tool provides valuable visual insights into abdominal wall dynamics, supporting surgical decision-making. However, it should be used as an adjunct rather than a standalone guide.

**Conclusions** This study presents a biomechanical strategy for hernia repair and introduces a visualisation tool that enhances preoperative assessment. While early results are promising, the tool's evolving nature and its role as a visual aid should be considered when interpreting outcomes. Further research is needed to validate its broader clinical utility.

## Plain language summary

People with abdominal wall problems, such as hernias, often experience pain and may need repeated surgeries. Traditional repair methods do not always consider how the abdominal wall behaves under pressure. In this study, we explored an approach that uses two imaging scans, one taken at rest and another during a breathing technique that puts the abdomen under strain. Comparing these scans shows how the abdominal wall moves and stretches, helping surgeons understand its biomechanics before surgery. Using this method, 31 patients were treated, and none reported pain or hernia recurrence after three years. These results suggest that incorporating detailed visual information into surgical planning may improve outcomes and make repairs safer and more effective.

Incisional hernia repair is often associated with chronic pain and high recurrence rates of 22–32%[1–3]. This is potentially due to an insufficient mechanical strength at the mesh-tissue interface. Despite the availability of various mesh types[4], surgical techniques[5], and fixation methods, each with its own advantages and disadvantages[1,6], the success of the repair largely depends on the size and location of the mesh used[7,8]. If the mesh is too small, it can lead to postoperative complications.

Preoperative manual assessment is commonly used to evaluate hernias and determine the risks of postoperative complications[9,10]. However, this process is time-consuming, challenging, and observer-dependent[11,12].

[1]Department of Materials Physics, Research School of Physics, The Australian National University, Acton, ACT, Australia. [2]Engineering Mathematics and Computing Lab (EMCL), Interdisciplinary Center for Scientific Computing (IWR), Heidelberg University, Heidelberg, Germany. [3]Data Mining and Uncertainty Quantification (DMQ), Heidelberg Institute for Theoretical Studies (HITS), Heidelberg, Germany. [4]Forschungscampus STIMULATE, Department of Fluid Dynamics and Technical Flows, Otto von Guericke University Magdeburg, Magdeburg, Germany. [5]Departement Chirurgie, Kantonsspital Graubünden, Walenstadt, Switzerland. [6]General and Visceral Surgery, Municipal Hospital Pirmasens, Pirmasens, Germany. [7]Radiological Center, Eberbach, Germany. [8]Diagnostic and Interventional Radiology, Heidelberg University Hospital, Heidelberg, Germany. [9]Translational Lung Research Center (TLRC) Heidelberg, German Center for Lung Research (DZL), Heidelberg, Germany. [10]General and Visceral Surgery, GRN Hospital, Eberbach, Germany. [11]General, Visceral and Transplantation Surgery, Heidelberg University Hospital, Heidelberg, Germany. [12]These authors contributed equally: Philipp D. Lösel, Jacob J. Relle. ✉e-mail: philipp.david.loesel@gmail.com

Customising the mesh to the unstable area of the abdominal wall, while accounting for cyclical loading, is crucial for achieving a durable and biomechanically stable repair[7,13]. It is important to note that the unstable area is often larger than the hernial orifice, and the hernia opening may not be centred within this area (Figs. 1 and 2). Using a mesh that only overlaps the defect area with a fixed margin can create weak spots, increasing the risk of recurrence (Fig. 1). Unfortunately, conventional approaches do not consider this factor, and there are currently no robust, fast, and automated preoperative evaluation techniques for hernias.

While pore size and mesh weight are considered critical factors, mesh size is typically selected to cover the defect area with a 5–7 cm overlap[14]. However, insufficient fixation of the mesh in unstable areas can lead to mesh sliding, shakedown, or ratcheting due to cyclic loading from external forces such as coughing or jumping. These forces transfer energy into the elastic-

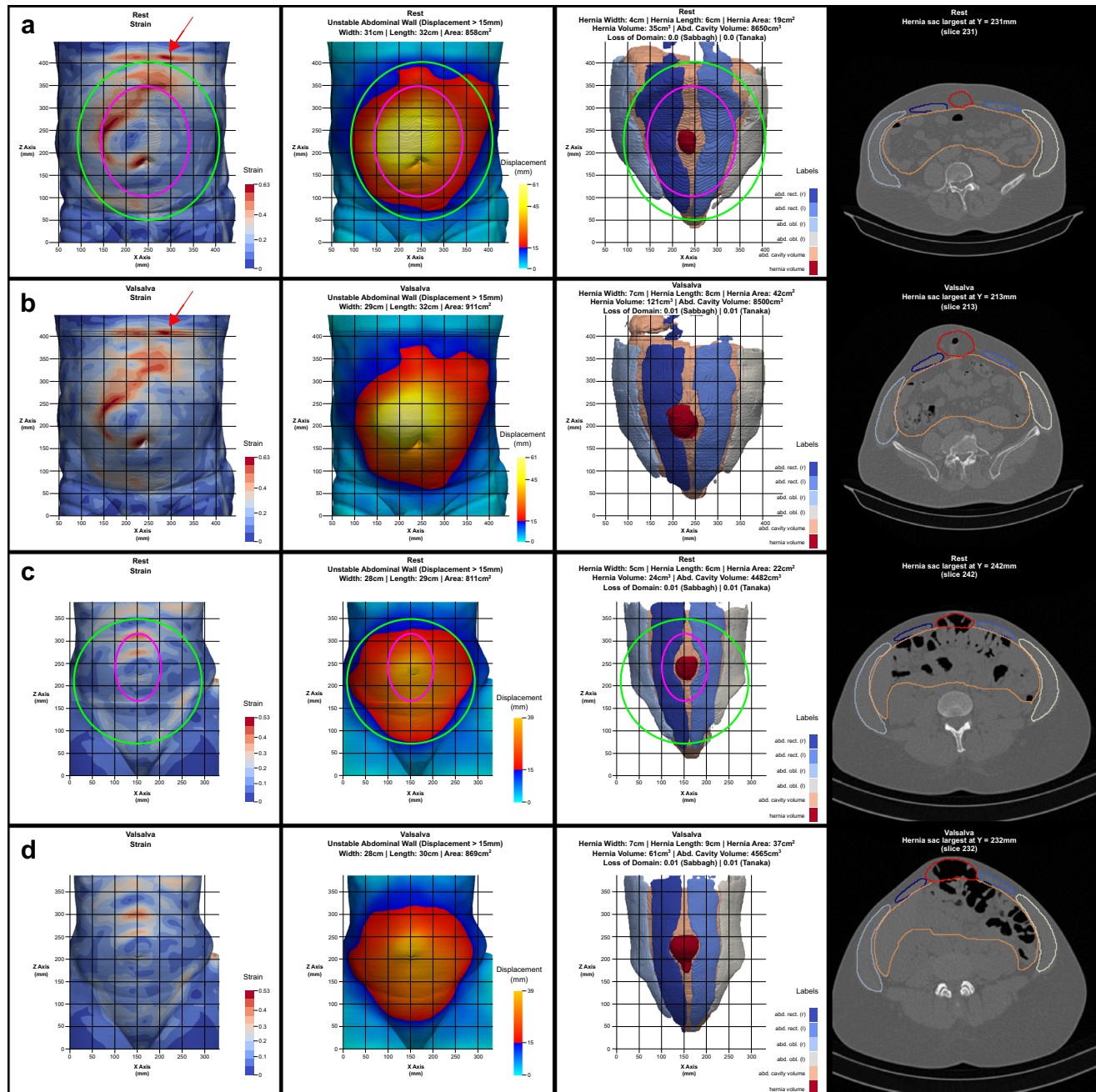

**Fig. 1 | HEDI results for two patients. a,b** HEDI result of a male patient with an intraoperatively measured defect area of 220 cm², a mesh size of 1060 cm² used, and an unstable abdominal wall of 858 cm² detected with HEDI. **c,d** HEDI result of a female patient with an intraoperatively measured defect area of 35 cm², a mesh size of 401 cm² used, and an unstable abdominal wall of 811 cm² detected with HEDI. Each result includes images of the patient at rest (**a,c**) and during the Valsalva manoeuvre (**b,d**), displaying strain, displacement with the unstable abdominal wall, automatic segmentation, and CT cross-sections (from left to right). The segmentation includes the volume of the abdominal cavity (beige), the muscle structures of the rectus (middle) and three-layered lateral muscles (shades of blue), and the hernia volume (red). Magenta ellipses illustrate a mesh covering only the defect area with a fixed overlap, while green ellipses illustrate a mesh covering the entire unstable area and strain hotspots. The strain hotspots in the first patient (**a,b**) at the top (indicated by red arrows) are caused by artificial shifts in the image data. The first patient (**a,b**) was not among the 35 manually annotated patients used for training and validating the automatic segmentation. In contrast, data from the second patient (**c,d**) were included in the training set.

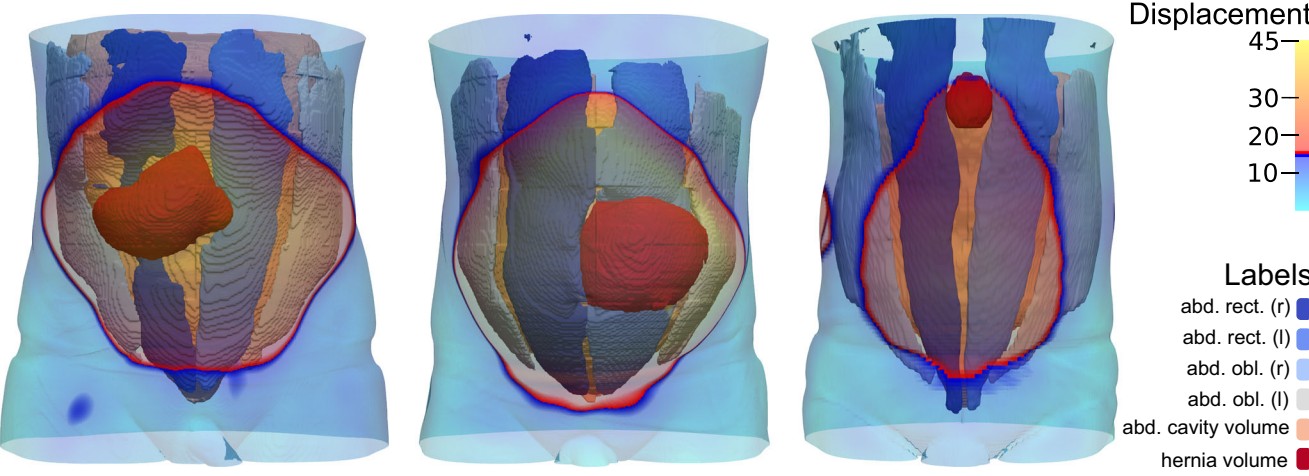

**Fig. 2 | Overlay of the unstable abdominal wall on the segmentation results.** Asymmetrical hernia openings within the unstable area of the abdominal wall highlight the need for individual consideration and customised repair design, as fixed overlaps are insufficient.

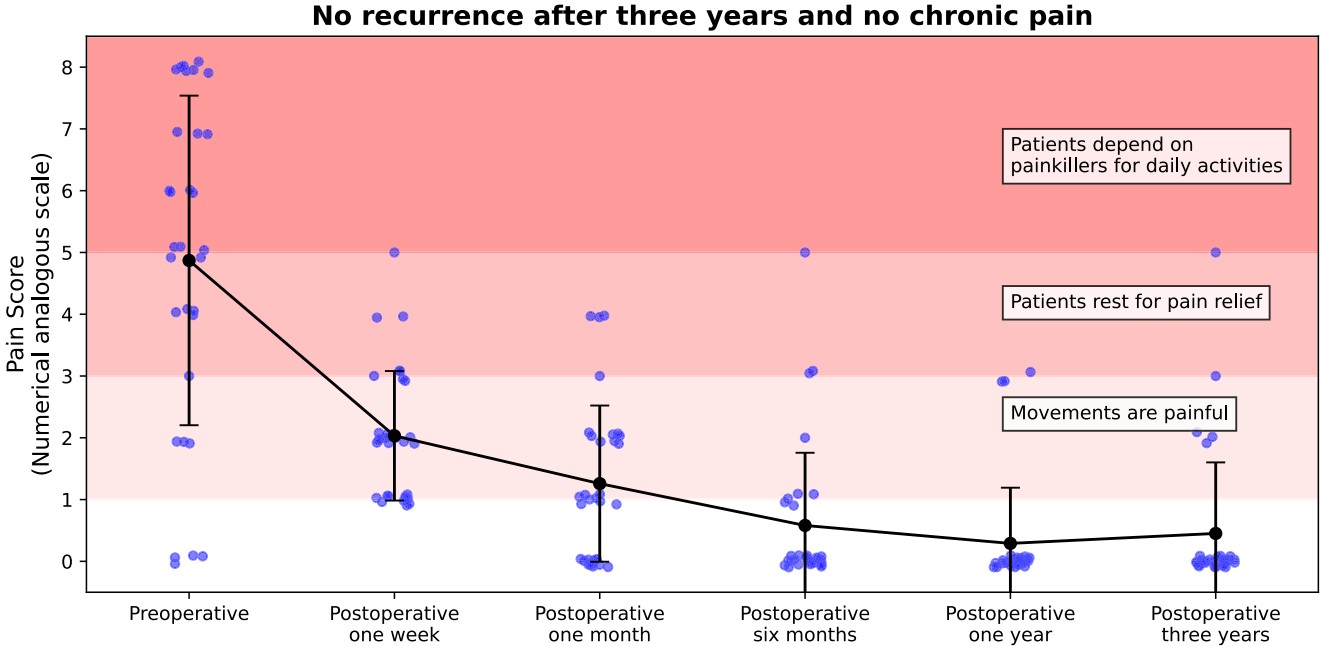

**Fig. 3 | Pain score of patients with incisional hernia repair.** Follow-up data from 31 patients is presented, with pain scores recorded on a numeric analogue scale of 0 to 10 before surgery and at increasing intervals up to three years after surgery. Mean pain scores are shown with error bars indicating standard deviation. In our clinical application, all 31 patients remained pain-free, and showed no evidence of hernia recurrence after three years of follow-up. In this study, "pain-free" refers to the absence of pain or the presence of minor discomfort that can be relieved by rest, without reliance on painkillers. Individual data points are shown with slight horizontal and vertical jitter ( ± 0.1) solely to enhance visual clarity when multiple values occur at the same time point; this jitter does not represent variation in time or pain scores.

plastic composite of the integrated mesh and surrounding soft tissue, which can ultimately result in mesh failure[15,16]. In our experience, to provide durable support and promote healing, the mesh must a) cover the entire unstable abdominal wall area and all strain hotspots, and b) be securely fixed in areas of low displacement and distortion. The mesh-to-defect area ratio is a critical factor in biomechanical approaches and has recently been recognised in updated guidelines (Chapter 14)[17]. Within the GRIP concept[13], a lower ratio is associated with an increased risk of recurrence, particularly in elastic tissue. Tissue elasticity can be assessed through tissue distension, and greater elasticity, indicated by larger displacement under pulse loading, requires a larger mesh-to-defect area ratio or reinforced fixation for a successful repair. However, manual evaluation of CT scans is time-consuming and can impede preoperative assessment.

To address these issues, we developed HEDI (Hernia Evaluation, Detection, and Imaging), a tool that evaluates and visualises the unstable abdominal wall, hernia size, and hernia volume utilising abdominal computed tomography (CT) in combination with the Valsalva manoeuvre (Fig. 1). The Valsalva manoeuvre increases the size and volume of a hernia and induces strain, revealing weaknesses in the surrounding tissue[18]. In this study, we define the unstable abdominal wall as the area of displacement greater than 15 mm during the Valsalva manoeuvre. In this first clinical application, HEDI was used as a visualisation tool to complement standard procedures based on the GRIP concept[13]; all 31 patients with available follow-up data remained pain-free and showed no evidence of hernia recurrence after three years (Fig. 3).

## Methods

### Statistics and reproducibility

Statistical analyses were performed to summarise patient characteristics and evaluate outcomes. Continuous variables are reported as mean ± standard deviation, and categorical variables as counts and percentages. No formal hypothesis testing was conducted, as the study was primarily descriptive.

The study included a total of 31 patients for clinical outcome evaluation and 141 patients for tool development. Each patient represented one independent replicate, as measurements were taken at the individual level. For imaging-based analyses, two CT scans per patient (at rest and during Valsalva manoeuvre) were acquired and processed as paired data. Segmentation and registration steps were evaluated against manually created reference labels in small subsets to confirm accuracy.

All analyses were performed using Python (version 3.10) and standard scientific libraries. Details of image processing and biomechanical calculations are provided in the following sections. The reproducibility of the workflow depends on consistent CT acquisition parameters and adherence to the described registration and segmentation protocols.

### Automatic segmentation

CT scans from 35 patients (ten women and 25 men, including 22 from the follow-up group) during shallow expiration and forced Valsalva manoeuvre were manually segmented twice by a clinical expert using Fiji[19], resulting in 140 annotated CT scans (Fig. 1c, d). For evaluation, Biomedisa's deep neural network was trained on image and label data from 21 patients using its standard configuration (version 24.5.23)[20]. The network was validated on seven patients during training and tested on seven patients after training. Data augmentation techniques, including horizontal flipping and random rotations between ±5 degrees, improved the Dice score for hernia volume segmentation by 15%. The neural network achieved average Dice scores of $0.93 \pm 0.01$ for abdominal cavity volume, $0.79 \pm 0.04$ for lateral muscle structure, $0.78 \pm 0.04$ for abdominal muscles, and $0.55 \pm 0.22$ for hernia volume, with mean absolute volume errors of $365 \pm 283$ mm$^3$, $71 \pm 56$ mm$^3$, $27 \pm 20$ mm$^3$, and $117 \pm 106$ mm$^3$, respectively. Hernia volume segmentation performed better on the CT scans during the Valsalva manoeuvre (Dice score of 0.64) compared to the CT scans at rest (Dice score of 0.49). For optimal performance, the network ultimately used in production was trained on 28 patients, including the seven validation patients, and validated on the seven test patients during training. This approach achieved an average Dice score improvement of 1.8% across all segments and 5.6% for hernia volume compared to the evaluation network, demonstrating the significant benefit of using more training data. The lower accuracy in hernia volume segmentation results from the substantial variability in hernia characteristics, including size, location, and internal structure, as well as the challenge of accurately delineating it from the surrounding abdominal region in training and test data. We expect that expanding the training dataset will improve the segmentation performance in the future. As a precaution, HEDI users should visually assess hernia volume segmentation before using the measured statistics for post-processing or decision-making.

### Symmetric diffeomorphic registration

The symmetric diffeomorphic registration was performed using the DIPY 1.7.0 package in Python[21], following a previously described method[22]. Image data was resampled to a voxel size of $1 \times 1 \times 1$ mm$^3$ and downscaled by a factor of 3 to speed up calculations. This factor balanced accuracy and computation time but can be adjusted individually when using HEDI. The two CT scans, taken at rest and during Valsalva, were converted into abdominal masks with the patient table removed using a threshold, resulting in a body outline. A single three-dimensional displacement field was then calculated, transforming the rest mask into the Valsalva mask using symmetric diffeomorphic registration[23]. The resulting vectors indicate the movement from a source voxel at rest to its corresponding location during the Valsalva manoeuvre. Both masks were converted into surface meshes and colour-coded based on the magnitude of the displacement vectors connecting each pair of points so that each point reflects the same deformation relative to the rest state.

### Evaluation of registration

The displacement calculated with HEDI was evaluated using data from three patients who underwent clinical examination at Heidelberg University Hospital, representing small, medium, and large displacements. For each patient, electrodes were placed on a regular grid with 5 cm spacing (Fig. 6), for a total of 30, 30, and 35 electrodes, respectively. These electrodes were manually identified in the image data and used as landmarks. The distances between instances at rest and Valsalva were measured and compared to the corresponding displacement values from HEDI. The average absolute errors were $1.6 \pm 1.6$ mm, and the normalised errors with respect to the maximum displacement of the electrodes averaged $4.6 \pm 4.3\%$. Given an average pixel size of $0.81 \times 0.81$ mm$^2$ and a slice thickness of 1 or 2 mm, these errors fall within the expected range of human error for landmark placement. It is important to note that using electrodes can make the registration process more robust and may therefore be considered in clinical practice.

### Handling inapplicable CT scans

Proper execution and reconstruction of CT scans are critical for accurate symmetric diffeomorphic registration. Maintaining the same patient table position for both acquisitions is essential. In obese patients with large hernias, the displacement of the abdominal wall may reach the edge of the field of view, which should be considered when planning the scan. Both CT scans should start and end on the same abdominal section for each patient, with identical slice number, thickness, increment, and field of view. Missing slices or differences in scaling (Fig. 5d) or shifts (Fig. 5c) between corresponding slices result in inaccurate displacement calculations. Truncation of CT scans (Fig. 5a) can also cause errors. To prevent scanning errors, any shift or scaling applied to one scan must also be applied to the other. Additionally, objects or body parts, such as arms (Fig. 5b), in the field of view can alter the surface and skew the calculated displacements.

### Ethics statement

The studies were reviewed and approved by the Ethics Committee of the Heidelberg University vote S-522/2020. The patients/participants provided their written informed consent to participate in this experimental study using non-certified procedures and software. The results were applied clinically by fully qualified, board certified surgeons using routine clinical procedures adapted to dimensionless measures of the stability towards dynamic intermittent strain.

## Results

### Valsalva manoeuvre, segmentation, and registration of the unstable abdominal wall

The development of HEDI involved 141 patients (67 women and 74 men) with an average age of $62 \pm 13$ ( ± standard deviation) years. All patients underwent two consecutive CT scans: one in tidal expiration (at rest) and another during a forced Valsalva manoeuvre, using either the Siemens SOMATOM Emotion 16 or Siemens SOMATOM Force CT scanner (Figs. 1 and 4). All patients were evaluated manually using the GRIP concept, with HEDI serving as an additional visualisation aid[13]. HEDI was not applicable in seven patients due to missing scans at rest or during the Valsalva manoeuvre, or different slice thicknesses between both scans. Additionally, 20 patients had flawed results due to scanning errors, such as different scaling, shift, or truncation of the abdomen (see "Handling inapplicable CT scans", Fig. 5). For the remaining 114 patients, the peak kilovoltage (kVp) ranged from 80 to 130, with an average of $109 \pm 5$ kVp. The exposure times ranged from 500 to 600 ms, with an average of $593 \pm 26$ ms. The slices were 1 or 5 mm thick, and the pixel spacing averaged $0.85 \pm 0.1$ mm in the X and Y directions. In both cases, Biomedisa's deep learning module[20,24] was used to segment the four muscle regions, the abdominal cavity volume, and the hernia volume (see "Automatic segmentation", Figs. 1 and 2). Volumes and loss of domain ratios

**Fig. 4 | Flow chart of the HEDI workflow.** From top to bottom: CT Scans at rest and during the Valsalva manoeuvre, generated masks of the labels and body outline, magnitude of the displacement field, and HEDI results with segmentation and displacement values mapped on the body surfaces.

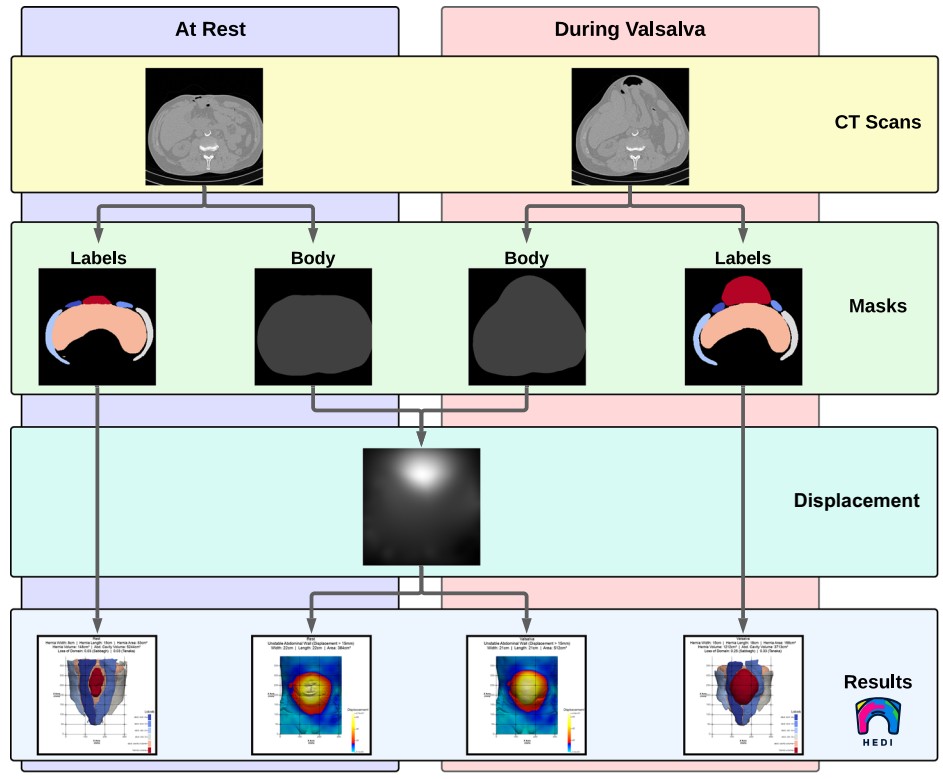

**Fig. 5 | Examples of inapplicable CT scans.** **a** Truncation of the content. **b** Objects such as arms included in the CT scan (white arrow). **c** Shift between CT scans at rest and during the Valsalva manoeuvre. **d** Different scaling between CT scans at rest and during the Valsalva manoeuvre.

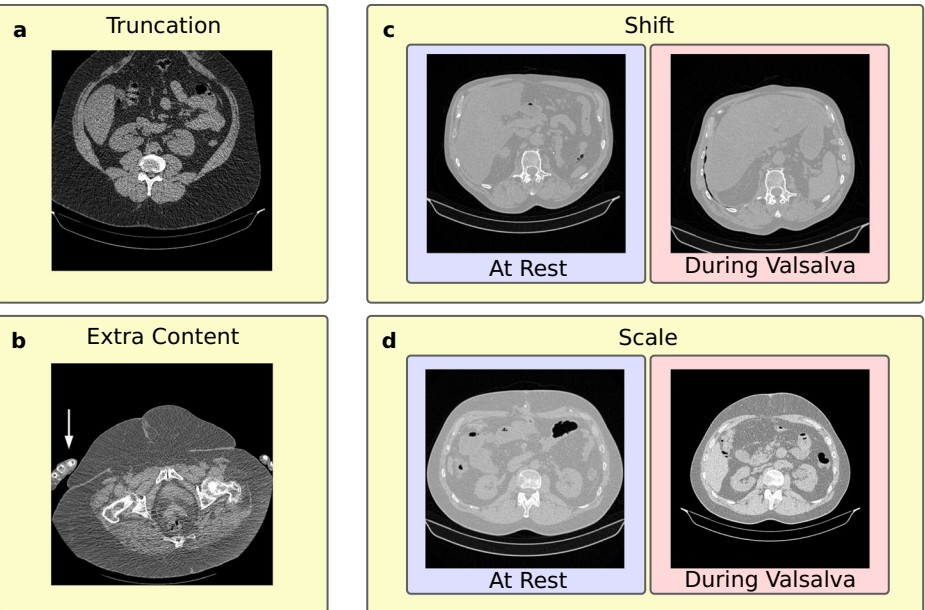

were calculated, and masks of the whole body were generated by thresholding the CT scans. These masks were used to determine the displacement field using symmetric diffeomorphic registration between the CT scans at rest and during the Valsalva manoeuvre (see "Symmetric diffeomorphic registration", Figs. 1, 2, and 6)[23]. For both instances, a 3D model of the abdominal surface was created, where areas with displacement greater than 15 mm were coloured in red, yellow, and white, while areas with displacement less than 15 mm were coloured in cyan and blue. Additionally, the Green-Lagrange strain tensor was calculated based on the displacement field. The magnitude of the displacement field and the maximum principal strain of each local strain tensor were visualised using ParaView 5.11 and annotated with hernia-related characteristics (Fig. 1).

HEDI integrates all methods into a single tool. Figure 4 illustrates the HEDI workflow. The final HEDI result is obtained by arranging screenshots of the individual steps alongside tomographic slices that display the largest area of the hernia sac (Fig. 1).

## Clinical application

Between March 2019 and July 2020, 31 patients (13 women and 18 men, average age of 58 ± 12 years) of this study underwent surgery considering both the preoperative evaluation using HEDI and the GRIP concept[13], with three-year follow-up data available. The respective hernia related sizes were determined manually (at least by three observers) and with HEDI. Out of these 31 patients, 18 had primary repairs, and 13 had recurrent repairs, with a total of

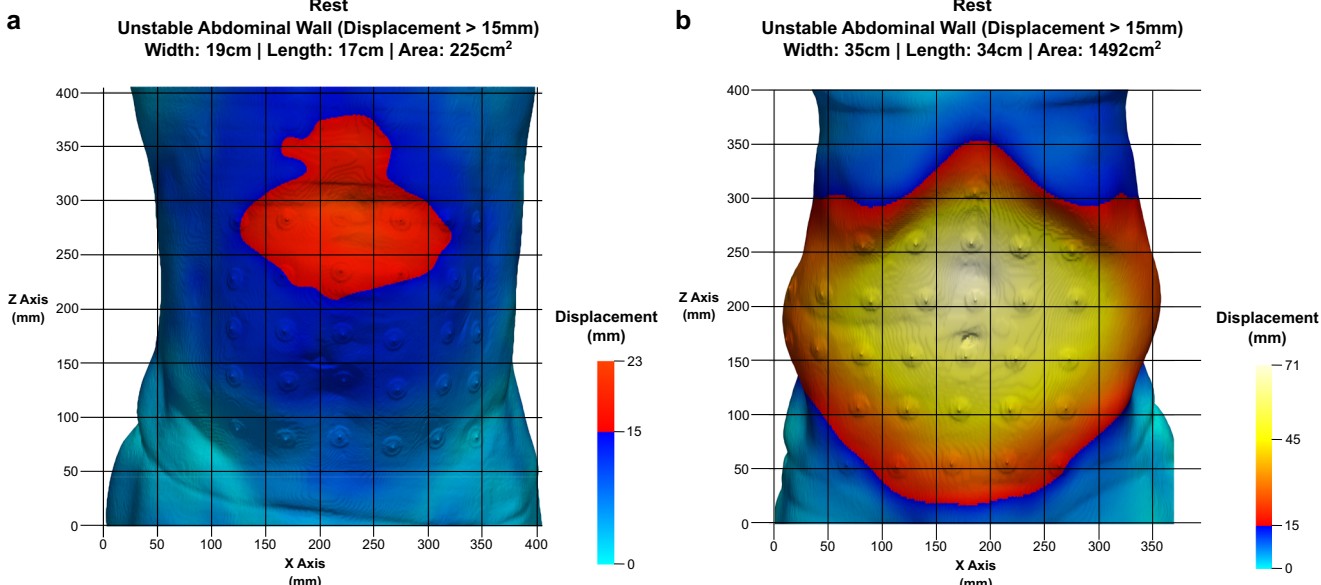

**Fig. 6 | HEDI results illustrating electrode use for evaluating the registration process. a** Patient at rest with a small maximum displacement of 23 mm and an unstable abdominal wall area of 225 cm². **b** Patient at rest with a large maximum displacement of 71 mm and an unstable abdominal wall area of 1492 cm². Both patients had 30 electrodes placed on the abdominal surface in a 5 cm spaced grid. Comparing manual measurements of electrode displacement with HEDI results showed an average absolute error of 1.6 mm and averaged normalised error with respect to the maximum displacement of the electrodes of 4.6%.

55 risk factors for recurrence[25]. Six patients were immunosuppressed, and four were transplant recipients. Twenty-three patients had incisional hernias wider than 10 cm, and eight had hernias between 4 and 10 cm in width. The average clinically assessed defect area was $210 \pm 136$ cm² (median: 181 cm², average width: 13 cm, average length: 19 cm). The average unstable abdominal wall area determined by HEDI was $610 \pm 488$ cm² (median: 470 cm²). The average computation time was 5 minutes 32 seconds using an Intel Core i7-13700K Processor and an NVIDIA GeForce RTX 4080. In all cases, a Dynamesh CICAT hernia mesh was used for repair, placed in a retromuscular, pre-peritoneal position. The average mesh size used was $1009 \pm 376$ cm² (median: 1060 cm², width range: 15–49 cm, length range: 30–49 cm). The minimum mesh extension over the hernia orifice averaged at $6.2 \pm 1.8$ cm (median: 6 cm), but varied significantly depending on the hernia orifice's location within the unstable abdominal wall (Figs. 1 and 2). A posterior release was necessary in 29 cases. Three cases required a sandwich repair (two with a concomitant parastomal hernia and one for a 491 cm² opening with a 93% loss of domain). The mesh-defect area ratio was $6 \pm 2.5$ (median: 5.5). Strain values derived from HEDI were visualised for each patient and used as input parameters within the GRIP framework[13]. Areas of elevated strain ("hotspots") were identified and specifically addressed with targeted fixation elements (average number of points: $127 \pm 69$, median: 115). The GRIP concept assesses repair quality based on characteristics such as mesh type, size, and fixation points. Because different combinations can achieve equivalent biomechanical sufficiency, GRIP recommendations are not unique. For example, a large mesh with few fixations may be considered equivalent to a smaller mesh with multiple fixation points. When several GRIP-compliant options were available, the preferred approach was the one that minimised operative time. Pain scores were collected from all patients before surgery and at increasing intervals up to three years after surgery. All 31 patients reported no pain and showed no evidence of hernia recurrence after three years of follow-up (Fig. 3). In this study, "pain-free" refers to the absence of pain or the presence of minor discomfort that can be relieved by rest, without reliance on painkillers.

## Discussion

The study demonstrates how HEDI can enhance preoperative evaluation in incisional hernia repair by providing a more detailed understanding of the biomechanical support required to achieve a stable repair.

Importantly, the clinical application of HEDI took place in parallel with its development, meaning that the visual outputs and processing steps of the tool evolved over time. As such, the results presented here reflect a dynamic integration of HEDI into surgical planning, and not a standardised, retrospective application of a finalised tool. Rather than serving as a deterministic decision-maker, HEDI provided surgeons with an additional source of biomechanical insight that complemented, but did not replace, their clinical expertise and judgement.

From a surgical perspective, HEDI enables tailored mesh fixation based on individual patient characteristics.

First and foremost, conventional approaches often determine mesh size and placement based solely on the hernia opening with a fixed overlap, represented by magenta ellipses in Fig. 1. However, our results emphasise the need for a substantially larger mesh to effectively cover the patient's entire unstable abdominal wall and strain hotspots, as indicated by green ellipses in Fig. 1.

Secondly, it's crucial to recognise that the hernia opening may not necessarily align with the centre of the unstable abdominal wall, as illustrated in Fig. 2. Relying solely on hernia opening coverage is insufficient, as this could position the mesh in unstable regions, making it vulnerable to destabilisation over time, particularly under cyclic loading conditions like coughing or jumping.

Lastly, customising mesh placement and fixation elements is crucial. In regions experiencing high strain, increasing the number of fixation elements and favouring running sutures over single sutures are critical adjustments. When abdominal wall loading varies significantly, a uniform fixation approach, such as "single crown", may not be appropriate. Instead, localisation should consider specific local displacements and distortions, particularly in areas with the highest strain. Recognising the importance of energy dissipation, we frequently employ continuous running sutures as robust support along the steep tissue shifts highlighted in red in the leftmost images of Fig. 1.

In summary, reconstruction decisions were informed by the visual and quantitative insights provided by HEDI; however, the final surgical approach, including mesh placement, type, size, fixation method, and overlap, remained at the discretion of the operating surgeon. As HEDI served as an adjunct rather than a directive system, it is difficult to infer strict

decision rules from its use alone. Nevertheless, we attribute the observed reductions in pain and recurrence rates, at least in part, to the enhanced visualisation and biomechanical understanding HEDI offers during pre-operative planning.

Although HEDI offers a first step in visualising the local displacement and strain of abdominal wall structures, it should be noted that the surgeon's judgement remains crucial. In addition, the study's findings are limited by the small sample size and short follow-up period. Future research should consider larger sample sizes, longer follow-up periods, and additional biomechanical aspects and advanced imaging techniques that complement the 14 research questions already formulated[26].

Additionally, HEDI results are influenced by errors in the image acquisition process. If the image data at rest and during Valsalva are not properly aligned (see "Handling inapplicable CT scans", Fig. 5) or contain artificial shifts (Fig. 1a,b), this can lead to (locally) incorrect results, which must be interpreted with caution. Another limitation of this study is the absence of a formal negative control arm. Clinical validation could be pursued by comparing outcomes before and after Biomechanically Calculated Reconstruction (BCR)[8] training, evaluating results with and without the use of HEDI. Moreover, the study was restricted to patients with large hernias treated by only three surgeons, and further investigation is needed to determine the optimal value for the instability threshold. Given that the exact displacement of the abdominal wall at which the mesh becomes unstable is still uncertain, we chose a relatively low threshold in this study to ensure that the mesh covers all unstable areas.

Our choice of a fixed 15 mm displacement threshold was initially based on literature data[7], which demonstrated its efficacy in distinguishing herniated from healthy or IPOM-repaired abdominal walls. Jourdan et al. further reported a maximum displacement of $17.9 \pm 8.0$ mm in healthy abdominal walls[27], supporting our selection of 15 mm as a cutoff between healthy and herniated tissues. Moreover, if such distension is present, the hernia mesh must be able to accommodate the resulting elastic deformation of the abdominal wall, and it is likely that its configuration becomes irreversibly altered once a certain threshold is exceeded. The precise value of this threshold, however, requires further investigation. We opted for an absolute value due to its simplicity and ease of comprehension. However, we acknowledge that the exact limit may vary among patients due to differences in abdominal wall characteristics, (e.g., tissue texture and elasticity), anatomical factors (e.g., muscle or sheath thickness), and physiological conditions, such as elevated intra-abdominal pressure (IAP). Consequently, we strongly encourage further investigations in this area to search for a relative, patient-specific, threshold. To account for such variability, HEDI offers flexibility by allowing users to set their own threshold based on their understanding of the individual patient's anatomy when starting the programme.

Additionally, a strain-based criterion could provide a more normalised and individualised assessment. However, our current focus has been on displacement because it is simple, clinically intuitive, and easily verifiable in practice. Surgeons can directly observe where large displacements occur under load and correlate these findings with dynamic CT (Valsalva manoeuvre) and HEDI output. By contrast, pre-calculated strain values cannot be readily verified at the patient's bedside.

Notably, while our primary focus is on large hernias, our smaller patient subgroup with available follow-up data outperforms reported literature statistics, including chronic pain and high recurrence rates across all hernia types, regardless of their size. Therefore, we are confident that our approach will potentially yield positive results when applied to a larger cohort. However, to further solidify our findings and establish a proper control group, we plan to employ propensity score matching (PSM) with the HERNIAMED registry once a larger cohort becomes available, consistent with our previous approach[28].

HEDI's application demonstrated feasibility and calculation reproducibility when applied to the CT data of our 141 patients, except for 27 cases affected by technical issues during data acquisition, as detailed in the "Handling inapplicable CT Scans" section of the Methods. During the development of HEDI, we compared different registration algorithms, with a specific focus on those capable of accommodating significant deformations and proven success in medical applications. Two methods were evaluated: one utilising B-splines[22] and the other employing symmetric diffeomorphic registration[21]. Both approaches yielded similar and comparable results in our dataset. The ultimate decision to use symmetric diffeomorphic registration from the DIPY package was based on both performance and licensing considerations.

However, assessing intra-patient variability remains challenging due to the limited frequency of patient scans, necessitated by the need to minimise radiation exposure. The robustness of unstable abdominal wall detection under repetition requires further investigation, ideally utilising non-invasive techniques such as MRI or laser scanning.

## Conclusion

HEDI enables visualisation of abdominal wall strain, displacement, and distortion under load, offering a patient-specific assessment of the biomechanical support required for a durable repair. While all patients in our cohort remained pain-free and experienced no recurrence, these encouraging outcomes must be interpreted with caution. HEDI was used alongside the GRIP concept during its iterative development, and the extent of its direct contribution to clinical results cannot be precisely quantified. Rather than serving as a prescriptive tool, HEDI functioned as a practical and well-accepted adjunct to surgical decision-making, allowing rapid evaluation of hernia size, volume, and abdominal wall instability from standard CT scans. This study demonstrates the feasibility of integrating biomechanical imaging into routine clinical workflows and highlights its potential value in supporting individualised hernia repair strategies. However, careful assessment of abdominal wall instability is essential to confirm clinical suspicion, particularly in small hernia defects, and to avoid both over- and under-treatment in critical cases.

## Data availability

Patients in this study did not provide consent for their data to be shared publicly. Therefore, tomographic imaging data are not publicly available. Additional data may be obtained from the corresponding author upon reasonable request, subject to review and approval by the Ethics Committee of the Medical Faculty of Heidelberg University, Germany. Available data include anonymised volumetric images, segmentation results, displacement fields, and individual data points underlying averaged results. An anonymised test dataset for code testing is provided at https://biomedisa.info/media/test_patient.zip. Requests are typically addressed within 30 days. The raw pain score data from 31 patients undergoing incisional hernia repair, which form the basis of Fig. 3, are provided as Supplementary Data with this manuscript.

## Code availability

The source code is available through the HEDI and Biomedisa open-source projects. It has been developed and tested on Windows 10 and Ubuntu 22.04 LTS. The code can be accessed at https://github.com/biomedisa/hernia-repair/ and is archived under DOI: 10.5281/zenodo.17705516[29]. Installation should follow the instructions provided in the repository.

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

## Acknowledgements

We acknowledge the support by the Heidelberg Foundation of Surgery (grant numbers 2018/215, 2019/288, 2020/376, and 2021/444), and the Informatics for Life project funded by the Klaus Tschira Foundation. P.D.L. has received funding from the Australian Research Council via the ARC Training Centre for Multiscale 3D Imaging, Modelling, and Manufacturing (M3D Innovation, project IC 180100008).

## Author contributions

P.D.L., J.J.R., R.N., V.H., and F.K. conceived and designed the study. P.D.L. and J.J.R. developed HEDI and carried out the evaluation. S.V. developed registration and strain tensor approach. R.R. segmented the training data and analysed the data. J.G. and M.O.W. performed original CT scans. F.K., T.L., and R.N. evaluated HEDI results, carried out hernia repair and follow-up evaluation. P.D.L. and F.K. supervised the project. P.D.L. and J.J.R. wrote the first draft of the manuscript. All authors contributed to the writing and discussion.

## Competing interests

The authors declare no competing interests.
