## [Transparent Peer Review file · Communications Medicine]

Clinical Application of HEDI for Biomechanical Evaluation and Visualisation in Incisional Hernia Repair

Corresponding Author: Dr Philipp Lösel

Version 0:

Reviewer comments:

Reviewer #1

(Remarks to the Author)

The work investigates the value of integrating biomechanical and geometrical data (by post processing CT images) for Incisional Hernia Repair. The authors present a method (with adequate literature references) and test its clinical value in a 62 ± 13 years old patient cohort, out of which 31 patients (age not reported) underwent surgery. All 31 patients remained pain-free, and showed no evidence of hernia recurrence after three years of follow-up. This reviewer has a number of serious concerns, related to the method and the conclusions drawn from this study. As a result, a proper assessment of the major claims made in this work, is impossible. Please see a list of major concerns below.

1. The definition of the “unstable area” by an absolute displacement of more than 15mm, needs further justification. Human anatomy varies largely, and at least a relative measure, i.e. relative to body size, or relative the mean abdominal wall displacement seems to be a better choice.
2. As a displacement always refers to two configurations, it is unclear how the abdominal wall displacement referring to “rest” or during the “Valsalva maneuver” was computed. As reported in the manuscript, “the mask at rest is transformed into the Valsalva mask by optimising an error measure using a symmetric diffeomorphic registration method. The registration produced a displacement field consisting of vectors pointing from a source pixel to its destination.” That’s perfectly understandable, but it is one single displacement (field) that can be calculated. Hence, the question: what is the rest-displacement and what is the Valsalva maneuver-displacement? In Fig.4 it seems that the same displacement is plotted twice, and then labeled differently?
3. In relation to point 2: What is the strain associated with “rest” and “Valsalva maneuver”?
4. Strain is calculated from the gradient of the displacement field and has clear physical meaning; normal strain reflects the change in length, and shear strain reflects the change of shape. As a matter of fact, a three-dimensional displacement field results in a six-dimensional strain field, which then prompts the question: which strain component, or combination of strain components, is shown in the figure/analyzed in this work. What does a strain level of 0.2 mean, for example? In addition, in relation to the selected frame of reference, many different strain definitions are known (engineering strain, Green-Lagrange-strain, Euler- Almansi strain, logarithmic strain,...). Which strain measure has been used in this work? This are not just semantic questions, but qualifies/disqualifies the applied method.
5. The statement “Although HEDI offers a first step in visualising the stress-strain relationship of abdominal wall structures” in the discussion section, is unclear. The method, as detailed in the manuscript measures some sort of kinematic quantity (denoted by strain but unlikely to be any physically meaningful strain), but it does, for sure, not consider stress by any means. Stress is a measure of tissue loading, not deformation, and the outlined method is not capable of computing stress.
6. As indicated by the authors, pain and recurrences plague appears in a number of patients after repair of an abdominal defect. However, it remain still unclear what the observation (all 31 patients operated on in this study remained pain-free, and showed no evidence of hernia recurrence after three years of follow-up) means. Any comparison with a proper control group would have helped to answer the question of the clinical benefit.
7. An elaboration on the implications of the relatively low DICE coefficient reported in section “automatic segmentation” would have been desired.
8. Statements/data concerning method validation would have been desired: the information shown in the color-coded images, how robust is it?

Reviewer #2

(Remarks to the Author)

Congratulations on a nice study. A few questions/comments?

- 1) this is nice technology and agree that we need more objective information on how we fix hernias. Do you have any sense on how this changed management, it appears that all of the stress was exactly where you think it would be in the middle of

the hernia defect. Can you describe any instances where this info changed what you did in surgery-use mesh not mesh, type of fixation or technique or how much overlap, myocutaneous flap?. I really like this technology but ideally we could use to help us make decisions in OR and think we need more information to do this
2) Discussion pretty short may want to add some more on this cool tecnology

Version 1:

Reviewer comments:

Reviewer #1

(Remarks to the Author)

Whilst some of the previous concerns have been clarified, main concern with the applied method stands, pls find a detailed response below.

Reviewer Comment C1.1. The definition of the “unstable area” by an absolute displacement of more than 15mm, needs further justification. Human anatomy varies largely, and at least a relative measure, i.e. relative to body size, or relative the mean abdominal wall displacement seems to be a better choice.

Response. Thank you for raising this crucial point. Our choice of a fixed 15 mm displacement threshold was initially based on literature data (Fig. 9 in Reference 19: Kallinowski et al. 2021), which demonstrated its efficacy in distinguishing between healthy and herniated or IPOM-repaired abdominal walls. Moreover, empirical evidence indicates that the mesh configuration becomes irreversibly altered when displacement exceeds approximately 15 mm. Additionally, we opted for an absolute value due to its simplicity and ease of comprehension. However, we acknowledge the potential for variability in this threshold among patients due to differences in abdominal wall characteristics, including tissue texture and elasticity. Consequently, we strongly encourage further investigations in this area. To account for such variability, HEDI offers flexibility by allowing users to set their own threshold based on their understanding of the individual patient’s anatomy when starting the program, as now mentioned in the discussion section.

As this is a severe limitation, it should be part of study limitations’ discussion.

Reviewer Comment C1.2. As a displacement always refers to two configurations, it is unclear how the abdominal wall displacement referring to “rest” or during the “Valsalva maneuver” was computed. As reported in the manuscript, “the mask at rest is transformed into the Valsalva mask by optimising an error measure using a symmetric diffeomorphic registration method. The registration produced a displacement field consisting of vectors pointing from a source pixel to its destination.” That’s perfectly understandable, but it is one single displacement (field) that can be calculated. Hence, the question: what is the rest-displacement and what is the Valsalva maneuver-displacement? In Fig.4 it seems that the same displacement is plotted twice, and then labeled differently?

Response. Thank you for highlighting this section, which could potentially lead to misunderstandings. Technically, the registration process yields a forward projection (from rest to Valsalva state) and a backward projection (from Valsalva to rest state). As a result, we distinguish between the configuration at rest and the configuration during the Valsalva maneuver. To prevent any misconceptions, we have modified the Method section, explicitly stating that two separate vector fields exist and describing the two different projections as follows:

“For each 2D slice, the mask at rest is transformed into the Valsalva mask by optimising an error measure using a symmetric diffeomorphic registration method. This process generates a map that facilitates the transition from the state at rest to the Valsalva state (forward projection) and the reverse transition from the Valsalva state back to the state at rest (backward projection). The resulting vectors either point from a source pixel at rest to its corresponding location during the Valsalva maneuver or in the opposite direction. The components of these Vectors pointing away from the centre of the mask defined the outward displacement during the Valsalva maneuver, while inward displacements were set to zero, and vice versa for the backward projection. Subsequently, both masks were then colour-coded based on the magnitude of the displacement vectors connecting each pair of points, resulting in two distinct 3D models showing the magnitude of translation between the two acquisition instances.”

Given this explanation, it is obvious that both vector fields represent the same displacement information – either recorded outwards or inwards. It then explains why the two fields plotted in Figure 1 are so similar. The fact that both fields are not exactly the same is simply:

1. the consequence of not using identical pixels in the displacement computation, and
2. some data has been erased (outwards pointing vectors in the back projection, and inwards pointing vectors in the forward projection).

Consequently, the difference between both fields is purely “numerical” and has no physical cause whatsoever.

Reviewer Comment C1.3. In relation to point 2: What is the strain associated with “rest” and “Valsalva maneuver”?

Response. The strain is calculated from the gradient of the corresponding displacement fields. As mentioned in our response to your Comment C1.2, we have one strain field for the configuration at rest and another for the Valsalva state. These are derived from the vectors of the forward and backward projections.

No, there is no physical meaning in the difference between both displacement fields, and as such among their spatial gradients either. See also the response above.

Reviewer Comment C1.4. Strain is calculated from the gradient of the displacement field and has clear physical meaning; normal strain reflects the change in length, and shear strain reflects the change of shape. As a matter of fact, a three-dimensional displacement field results in a six dimensional strain field, which then prompts the question: which strain component, or combination of strain components, is shown in the figure/analyzed in this work. What does a strain level of 0.2 mean, for example? In addition, in relation to the selected frame of reference, many different strain definitions are known (engineering strain, Green-Lagrange-strain, Euler- Almansi strain, logarithmic strain,...). Which strain measure has been used in this work? This are not just semantic questions, but qualifies/disqualifies the applied method.

Response. That's an excellent observation. The local strain is defined by nine components, six of which are independent. We calculated the Euler-Almansi strain Tensor based on the displacement field, following the methodology outlined in reference 18 (Abd-Elmoniem et al. 2008). This tensor is often reduced to a single local value to facilitate comparison with results from mechanical testing or for visualization purposes. In our study, we adopted the von Mises equation to compute a scalar field representing the strain. While this equation is typically associated with stress calculations, in this instance, we adapted it for strain analysis. For clarity and completeness, we have incorporated the following sentence into the Results section (lines 111-112):

"The local strain tensor was reduced to a single value equivalent to the von Mises equation for mechanical stress."

Ref 18 indeed worked with the Euler-Almansi strain, but interestingly called it 3-D Eulerian strain. However, it is essential that it is only the Euler-Almansi strain, given the displacement field stems from the forward projection. One CANNOT take the displacement field from the backward projection and put it into eq(4) of ref 18, i.e. the definition of the Euler-Almansi strain. The result would simply make no physical sense.

In addition, the computation of the average strain (application of the von Mises stress to the strain tensor) lacks physical meaning in the present context. One can easily show that a large strain value could in fact result from zero tangential strain in the skin. Instead using the maximum principal strain at the surface (in the the skin) would be a sensitive strain measure in the present context.

In addition, the way the displacement fields have been generated, i.e. morphing 2d slices, neglects any strain that would appear in the third (out-of-plane) direction.

In addition, the displacement vector that is used in strain definitions has to connect the reference and spatial configurations of a single material particle. As morphing cannot guarantee that the two connected pixels belong to the same material particle, the resulting displacement is questionable in the computation of a strain field.

As a consequence, and in addition to the statement made in response to C1.2, the present methodology is severely flawed, and it does not make any sense to discuss results from such an analysis.

Reviewer Comment C1.5. The statement "Although HEDI offers a first step in visualising the stress-strain relationship of abdominal wall structures" in the discussion section, is unclear. The method, as detailed in the manuscript measures some sort of kinematic quantity (denoted by strain but unlikely to be any physically meaningful strain), but it does, for sure, not consider stress by any means. Stress is a measure of tissue loading, not deformation, and the outlined method is not capable of computing stress.

Response. We fully agree with this comment. This statement is not justified based on our approach. The connection between strain and stress has not been investigated or visualized. We have revised the statement as follows: "Although HEDI offers a first step in visualising the local displacement and strain stress-strain relationship of abdominal wall structures, [...]"

Reviewer Comment C1.6. As indicated by the authors, pain and recurrences plague appears in a number of patients after repair of an abdominal defect. However, it remain still unclear what the observation (all 31 patients operated on in this study remained pain-free, and showed no evidence of hernia recurrence after three years of follow-up) means. Any comparison with a proper control group would have helped to answer the question of the clinical benefit.

Response. We appreciate your comment and acknowledge the importance of a robust control group to establish the superiority of the HEDI-based approach. The primary objective of this study is to introduce the HEDI tool, developed as part of our new approach, and demonstrate its feasibility in a cohort of 141 patients, with follow-up data available for 31 patients. Notably, while our primary focus is on large hernias, our smaller patient subgroup with available follow-up data outperforms reported literature statistics, including chronic pain and high recurrence rates (reported as 22-32%) across all hernia types, regardless of size. Therefore, we are confident that our approach will potentially yield positive results when applied to a larger cohort. However, to further solidify our findings and establish a proper control group, we plan to employ propensity score matching (PSM) with the HERNIAMED@ registry once we gather a larger cohort, as now mentioned in the discussion section. Reviewer

As this is a severe limitation, it should be part of study limitations' discussion.

Comment C1.7. An elaboration on the implications of the relatively low DICE coefficient reported in section “automatic segmentation” would have been desired.

Response. Thank you for this important comment. The lower accuracy observed in hernia volume segmentation results from the substantial variability in hernia characteristics, including differences in size, location, and internal structure, as well as the inherent challenge of accurately delineating it from the surrounding abdominal region. It's important to note that we identified a technical issue in the TensorFlow implementation for Multi-GPU, specifically related to Batch Normalization. In this issue, Batch Normalization was being performed on individual subbatches on each GPU rather than on the entire batch. To address this problem, we decided to use synchronized Batch Normalization layers (improvement from 0.45 to 0.49 for hernia volume). Furthermore, we found that data augmentation, specifically horizontally flipping and randomly rotating the images, had a positive impact on improving the Dice score for hernia segmentation (improvement from 0.49 to 0.58 for hernia volume). However, it's essential to acknowledge that the Dice coefficient for hernia volume segmentation remains relatively low. As a result, we recommend that users of HEDI exercise caution and visually assess the segmentation performance of the hernia volume. This visual validation step should be conducted before relying on the measured statistics of the hernia volume for any post-processing or decision-making purposes, as now mentioned in this section.

It would still be worth quantifying its meaning in term of error in volume measurements.

Reviewer Comment C1.8. Statements/data concerning method validation would have been desired: the information shown in the color-coded images, how robust is it?

Response. Thank you for your valuable comment. First, HEDI's application demonstrated feasibility and calculation reproducibility when applied to the CT data of our 141 patients, except for 27 special cases detailed in the 'Inapplicable CT Scans' section of the Methods. Second, during the development of HEDI, we compared different registration algorithms, with a specific focus on those capable of accommodating significant deformations and proven success in medical applications. Two methods were evaluated: one utilising B-splines (Reference 24: Voß et al. 2020) and the other employing symmetric diffeomorphic registration (Reference 23). Both approaches yielded similar and comparable results in our dataset. This consistency provided us with confidence in the algorithm's validity. The ultimate decision to use symmetric diffeomorphic registration from the DIPY package was based on both performance and licensing considerations. However, assessing intra-patient variability remains challenging due to the limited frequency of patient scans, necessitated by the need to minimise radiation exposure. The robustness of unstable abdominal wall detection under repetition requires further investigation, ideally utilising non-invasive techniques such as MRI or laser scanning, as now mentioned in the discussion.

Clearly, validation in patients (except cancer patients) is difficult. However, the system could be easily validated using phantoms, an exercise that would then provide valuable data to judge its robustness.

Reviewer #2

(Remarks to the Author)

Author's queries appropriately addressed

Version 2:

Reviewer comments:

Reviewer #1

(Remarks to the Author)

The authors provided some more information, which however, did not contribute much to the clarity of this work. Key points remain unanswered:

1. Given one displacement field one can only compute one strain field, and it is completely unclear why two strain fields are shown in Fig 1.
2. For some reason the authors decided to switch to the Green-Lagrange strain. However, the response to referee text always talks about the Euler-Almansi strain. What strain measure has been used in the plots?
3. How did the strain field influence the treatment decision?
4. What data provided by HEIDI has been used in the treatment decision and why should that be superior to conventional decision making? At the moment, no evidence justifies the first sentence in the discussion section.
5. Besides using HEIDI, the manuscript reports the use of the GRIP concept. What is it and how was it used?
6. The flow chart in Figure 4 misses the strain data – why?
7. Concerning the newly added evaluation of the registration process: What was the ground truth that allowed the authors to come up with a 1.6mm error in Figure 6? In addition, placing electrodes defines a fundamentally different (more robust) registration problem.

Reviewer #3

(Remarks to the Author)

The paper address issues of hernia repair. The authors propose the computational tool supporting hernia surgery and show the promising clinical outcomes. The patient evaluation is based on the displacement field of the abdominal wall during Valsalva maneuver. The concept is interesting and the paper reports present novel outcomes. However, in my opinion some aspects should be discussed/ described in more detail, mainly regarding assumption that instability occurs when displacement of the abdominal wall during Valsalva maneuver exceeds 15 mm.

- 1) Could you write the basis of the statement "Moreover, empirical evidence indicates that the mesh configuration becomes irreversibly altered when displacement exceeds approximately 15 mm." ? Is this clinical experience supported by some imaging? What happens to the mesh?
- 2) Could you comment on the limit referring to the healthy abdominal wall behavior. Jourdan et al. 2022 showed the displacement of abdominal wall of healthy abdominal wall during Valsalva maneuver.
Jourdan , et al. "Dynamic-MRI quantification of abdominal wall motion and deformation during breathing and muscular contraction." *Computer Methods and Programs in Biomedicine* 217 (2022): 106667
- 3) Could you please explain in more detail why exceeding given displacement value is treated as a reason to overlap given area?
- 4) I think that you should write in the text what is the reference state in the calculation of displacement and strain when you show each strain/displacement map. I suppose you change the reference when showing "rest" and non-rest maps. From the response to another reviewer, I understand the sores of difference. However, I think that all of this should be explained in the text to the readers.

Reviewer #4

(Remarks to the Author)

Congratulations for this study that introduces novel possibilities for tailoring mesh fixation to individual patient characteristics. Such approach appears to be the next brick needed to improve the clinical outcomes following hernia repair. The paper is well-written. However, there are also limitation which should be better explicated. Please find below my comments

- Introduction: Incisional hernia repair is often associated with chronic pain and high recurrence rates of 22-32%¹⁻³. [This is mostly due to an insufficient mechanical strength at the mesh-tissue interface.] Despite the availability of various mesh types⁴, surgical techniques⁵, and fixation methods, each with its own advantages and disadvantages^{1,6}, [the success of the repair largely depends on the size and location of the mesh used.] If the mesh is too small or [fixed under tension], it can lead to postoperative complications.
Please could you give references to justify the bracketed parts? If there are assumptions, they should be reformulated as such.
- Line 73: What exactly do you mean by "composite" material? Are you referring to the mesh and the surrounding soft tissue before or after mesh integration?
- Line 74: The criteria the authors mentioned are not intuitive. Are these criteria needed to prevent sliding or to provide overall a good stability of the repair? Why does a mesh need to be secured in a low displacement region? During a Rives technique for example, the mesh is secured in the rectus muscle and/or the posterior rectus sheath, which is likely the region of the AW with the highest mobility (in terms of antero-posterior displacement). Yet, this technique is probably the safest today.
- Line 78: Although it makes totally sense to normalize the results with respect to quality of the tissue, higher displacement under pulse loading may come from a higher elasticity of the tissue but also due to the anatomy (muscle/sheath thickness) or the physiology (high IAP).
- Line 88: How do you justify the relevance of using this 15-mm criteria identified on ex vivo porcine ex situ AW sample for living human patients? Given the inter-individual variability regarding the anatomy, tissue elasticity, IAP etc, using the same displacement-based criteria for all patients raises some questions. Why not using a strain-based criteria that would provide some normalization?
- Lines 137-140: Given the size of the meshes used, I am presuming a component separation technique was performed for some patients. Could you say more about this? Excepted the size of the mesh, was a CST performed for other reasons (e.g., too much tension in the posterior rectus sheath/loss of domain)?
- Line 162: Are you sure these strain hotspots are actual hotspots and are not derived from "side effects" during the registration/differentiation process. Fig 1a) 1b) show for example spots at the interface ribs/rectus muscle. Do the hotspots reveal a high-strain region or from a region with a high gradient between a low-motion region (ribs) and a high-motion region (muscle)? The strain field derived from a spatial differentiation of the displacement field which amplifies the noise already present in the displacement field. In other words, although a validation was performed on a continuous field (Evaluation of Registration) what level of validation does the image processing have (segmentation registration differentiation)?
- Fig 2 (right), the defect seems located in the most cranial region of the AW. Also, the unstable region detected by HEDI seems to show that the most cranial part of the defect should ne be covered. Yet it is known that an insufficient mesh overlap in that region due to an insufficient sub-xiphoid dissection may lead to recurrence as well. In that case, what does HEDI recommend? Please specify.
- Line 128: It is said that the HEDI and the GRIP concepts were used for this study. Based on the literature, the GRIP concept assesses the quality of the repair based on the repair characteristics (mesh type, size, number of fixations etc). Therefore, the recommendations provided by the GRIP concept are not unique. As different repair techniques could be

equivalent from a GRIP standpoint (e.g., large mesh without fixation vs. smaller mesh with many fixations points), could the author specify which strategy in this study was used?

- Although this study seems to provide good results in terms of recurrence rate, it also raises some questions about the potential overkill associated with this technique. Would the patient in Fig1a with a small defect have complications with a mesh in the retro rectus area providing a 5-cm overlap? Although CST offers advantages (e.g., decreased tension in the midline, use of large mesh), it is also associated with limitations (e.g., limited long-term impact, invasiveness) that should be explicated here.
- The absence of a “negative control” arm should be mentioned as a limitation of this study and a methodology providing a clinical validation of this approach should be provided.

Version 3:

Reviewer comments:

Reviewer #3

(Remarks to the Author)

The authors improved paper and answered my comments. I think that the paper could be published in the current form.

Reviewer #4

(Remarks to the Author)

Thank you for your answers. One suggestion:

R4C9: Line 128: It is said that the HEDI and the GRIP concepts were used for this study. Based on the literature, the GRIP concept assesses the quality of the repair based on the repair characteristics (mesh type, size, number of fixations etc). Therefore, the recommendations provided by the GRIP concept are not unique. As different repair techniques could be equivalent from a GRIP standpoint (e.g., large mesh without fixation vs. smaller mesh with many fixations points), could the author specify which strategy in this study was used?

AQR4C9: Thank you for the insightful comment. You are correct that the GRIP concept allows for different combinations of mesh type, size, and fixation strategy to achieve biomechanical sufficiency. When multiple GRIP-compliant options were available, the preferred strategy was the one that minimised operative time, provided it still achieved GRIP > CRIP.

Please specify this in the manuscript as it is an critical factor in my opinion.

Response to Referees

COMMSMED-23-0436

HEDI: First-Time Clinical Application and Results of a Biomechanical Evaluation and Visualisation Tool for Incisional Hernia Repair

Reviewers' comments:

Reviewer #1 (Remarks to the Author):

Reviewer Summary. The work investigates the value of integrating biomechanical and geometrical data (by post processing CT images) for Incisional Hernia Repair. The authors present a method (with adequate literature references) and test its clinical value in a 62 ± 13 years old patient cohort, out of which 31 patients (age not reported) underwent surgery. All 31 patients remained pain-free, and showed no evidence of hernia recurrence after three years of follow-up. This reviewer has a number of serious concerns, related to the method and the conclusions drawn from this study. As a result, a proper assessment of the major claims made in this work, is impossible. Please see a list of major concerns below.

Reviewer Comment C1.1. The definition of the “unstable area” by an absolute displacement of more than 15mm, needs further justification. Human anatomy varies largely, and at least a relative measure, i.e. relative to body size, or relative the mean abdominal wall displacement seems to be a better choice.

Response. Thank you for raising this crucial point. Our choice of a fixed 15 mm displacement threshold was initially based on literature data (Fig. 9 in Reference 19: Kallinowski et al. 2021), which demonstrated its efficacy in distinguishing between healthy and herniated or IPOM-repaired abdominal walls. Moreover, empirical evidence indicates that the mesh configuration becomes irreversibly altered when displacement exceeds approximately 15 mm. Additionally, we opted for an absolute value due to its simplicity and ease of comprehension. However, we acknowledge the potential for variability in this threshold among patients due to differences in abdominal wall characteristics, including tissue texture and elasticity. Consequently, we strongly encourage further investigations in this area. To account for such variability, HEDI offers flexibility by allowing users to set their own threshold based on their understanding of the individual patient’s anatomy when starting the program, as now mentioned in the discussion section.

Reviewer Comment C1.2. As a displacement always refers to two configurations, it is unclear how the abdominal wall displacement referring to “rest” or during the “Valsalva maneuver” was computed. As reported in the manuscript, “the mask at rest is transformed into the Valsalva mask by optimising an error measure using a symmetric diffeomorphic registration method. The registration produced a displacement field consisting of vectors pointing from a source pixel to its destination.” That’s perfectly understandable, but it is one single displacement (field) that can be calculated. Hence, the question: what is the rest-displacement and what is the Valsalva maneuver-displacement? In Fig.4 it seems that the same displacement is plotted twice, and then labeled differently?

Response. Thank you for highlighting this section, which could potentially lead to misunderstandings. Technically, the registration process yields a forward projection (from rest to Valsalva state) and a backward projection (from Valsalva to rest state). As a result, we distinguish between the configuration at rest and the configuration during the Valsalva maneuver. To prevent any misconceptions, we have modified the Method section, explicitly stating that two separate vector fields exist and describing the two different projections as follows:

“For each 2D slice, the mask at rest is transformed into the Valsalva mask by optimising an error measure using a symmetric diffeomorphic registration method. This process generates a map that facilitates the transition from the state at rest to the Valsalva state (forward projection) and the reverse transition from the Valsalva state back to the state at rest (backward projection). The ~~registration produced a displacement field consisting of~~ resulting vectors either pointing from a

source pixel at rest to its destination corresponding location during the Valsalva maneuver or in the opposite direction. ~~The components of these v~~ectors pointing away from the centre of the mask defined the outward displacement during the Valsalva maneuver, while inward displacements were set to zero, and vice versa for the backward projection inverse map. Subsequently, both masks were then colour-coded based on the magnitude of the displacement vectors connecting each pair of points, resulting in two distinct 3D models showing the magnitude of translation between the two acquisition instances.

Reviewer Comment C1.3. In relation to point 2: What is the strain associated with “rest” and “Valsalva maneuver”?

Response. The strain is calculated from the gradient of the corresponding displacement fields. As mentioned in our response to your Comment C1.2, we have one strain field for the configuration at rest and another for the Valsalva state. These are derived from the vectors of the forward and backward projections.

Reviewer Comment C1.4. Strain is calculated from the gradient of the displacement field and has clear physical meaning; normal strain reflects the change in length, and shear strain reflects the change of shape. As a matter of fact, a three-dimensional displacement field results in a six-dimensional strain field, which then prompts the question: which strain component, or combination of strain components, is shown in the figure/analyzed in this work. What does a strain level of 0.2 mean, for example? In addition, in relation to the selected frame of reference, many different strain definitions are known (engineering strain, Green-Lagrange-strain, Euler- Almansi strain, logarithmic strain,...). Which strain measure has been used in this work? This are not just semantic questions, but qualifies/disqualifies the applied method.

Response. That's an excellent observation. The local strain is defined by nine components, six of which are independent. We calculated the Euler-Almansi strain Tensor based on the displacement field, following the methodology outlined in reference 18 (Abd-Elmoniem et al. 2008). This tensor is often reduced to a single local value to facilitate comparison with results from mechanical testing or for visualization purposes. In our study, we adopted the von Mises equation to compute a scalar field representing the strain. While this equation is typically associated with stress calculations, in this instance, we adapted it for strain analysis. For clarity and completeness, we have incorporated the following sentence into the Results section (lines 111-112):

“The local strain tensor was reduced to a single value equivalent to the von Mises equation for mechanical stress.”

Reviewer Comment C1.5. The statement “Although HEDI offers a first step in visualising the stress-strain relationship of abdominal wall structures” in the discussion section, is unclear. The method, as detailed in the manuscript measures some sort of kinematic quantity (denoted by strain but unlikely to be any physically meaningful strain), but it does, for sure, not consider stress by any means. Stress is a measure of tissue loading, not deformation, and the outlined method is not capable of computing stress.

Response. We fully agree with this comment. This statement is not justified based on our approach. The connection between strain and stress has not been investigated or visualized. We have revised the statement as follows:

“Although HEDI offers a first step in visualising the local displacement and strain ~~stress-strain relationship~~ of abdominal wall structures, [...]”

Reviewer Comment C1.6. As indicated by the authors, pain and recurrences plague appears in a number of patients after repair of an abdominal defect. However, it remain still unclear what the observation (all 31 patients operated on in this study remained pain-free, and showed no evidence of

hernia recurrence after three years of follow-up) means. Any comparison with a proper control group would have helped to answer the question of the clinical benefit.

Response. We appreciate your comment and acknowledge the importance of a robust control group to establish the superiority of the HEDI-based approach. The primary objective of this study is to introduce the HEDI tool, developed as part of our new approach, and demonstrate its feasibility in a cohort of 141 patients, with follow-up data available for 31 patients.

Notably, while our primary focus is on large hernias, our smaller patient subgroup with available follow-up data outperforms reported literature statistics, including chronic pain and high recurrence rates (reported as 22-32%) across all hernia types, regardless of size. Therefore, we are confident that our approach will potentially yield positive results when applied to a larger cohort.

However, to further solidify our findings and establish a proper control group, we plan to employ propensity score matching (PSM) with the HERNIAMED[®] registry once we gather a larger cohort, as now mentioned in the discussion section.

Reviewer Comment C1.7. An elaboration on the implications of the relatively low DICE coefficient reported in section “automatic segmentation” would have been desired.

Response. Thank you for this important comment. The lower accuracy observed in hernia volume segmentation results from the substantial variability in hernia characteristics, including differences in size, location, and internal structure, as well as the inherent challenge of accurately delineating it from the surrounding abdominal region. It's important to note that we identified a technical issue in the TensorFlow implementation for Multi-GPU, specifically related to Batch Normalization. In this issue, Batch Normalization was being performed on individual subbatches on each GPU rather than on the entire batch. To address this problem, we decided to use synchronized Batch Normalization layers (improvement from 0.45 to 0.49 for hernia volume). Furthermore, we found that data augmentation, specifically horizontally flipping and randomly rotating the images, had a positive impact on improving the Dice score for hernia segmentation (improvement from 0.49 to 0.58 for hernia volume).

However, it's essential to acknowledge that the Dice coefficient for hernia volume segmentation remains relatively low. As a result, we recommend that users of HEDI exercise caution and visually assess the segmentation performance of the hernia volume. This visual validation step should be conducted before relying on the measured statistics of the hernia volume for any post-processing or decision-making purposes, as now mentioned in this section.

Reviewer Comment C1.8. Statements/data concerning method validation would have been desired: the information shown in the color-coded images, how robust is it?

Response. Thank you for your valuable comment. First, HEDI's application demonstrated feasibility and calculation reproducibility when applied to the CT data of our 141 patients, except for 27 special cases detailed in the 'Inapplicable CT Scans' section of the Methods.

Second, during the development of HEDI, we compared different registration algorithms, with a specific focus on those capable of accommodating significant deformations and proven success in medical applications. Two methods were evaluated: one utilising B-splines (Reference 24: Voß et al. 2020) and the other employing symmetric diffeomorphic registration (Reference 23). Both approaches yielded similar and comparable results in our dataset. This consistency provided us with confidence in the algorithm's validity. The ultimate decision to use symmetric diffeomorphic registration from the DIPY package was based on both performance and licensing considerations.

However, assessing intra-patient variability remains challenging due to the limited frequency of patient scans, necessitated by the need to minimise radiation exposure. The robustness of unstable abdominal wall detection under repetition requires further investigation, ideally utilising non-invasive techniques such as MRI or laser scanning, as now mentioned in the discussion.

Reviewer #2 (Remarks to the Author):

Reviewer Summary. Congratulations on a nice study. A few questions/comments? this is nice technology and agree that we need more objective information on how we fix hernias.

Reviewer Comment C2.1. Do you have any sense on how this changed management, it appears that all of the stress was exactly where you think it would be in the middle of the hernia defect. Can you describe any instances where this info changed what you did in surgery-use mesh not mesh, type of fixation or technique or how much overlap, myocutaneous flap?. I really like this technology but ideally we could use to help us make decisions in OR and think we need more information to do this

Response. Your comment is greatly appreciated, as it touches upon the central theme of our study - illustrating the impact of HEDI results on surgical decision-making. There are several critical points we wish to emphasize which we have added to the discussion:

“From a surgical perspective, HEDI introduces novel possibilities for tailoring mesh fixation to individual patient characteristics. First and foremost, conventional approaches often determine mesh size and placement based solely on the hernia opening with a fixed overlap, represented by the magenta rectangle in Fig. 1. However, our results emphasise the need for a substantially larger mesh to effectively cover the patient’s entire unstable abdominal wall, as indicated by the green rectangle in Fig. 1. Secondly, it’s crucial to recognize that the hernia opening may not necessarily align with the centre of the unstable abdominal wall, as illustrated in Fig. 2. Relying solely on hernia opening coverage is insufficient, as this could position the mesh in unstable regions, making it vulnerable to destabilisation over time, particularly under cyclic loading conditions like coughing or jumping. Lastly, customising mesh placement and fixation elements is crucial. In regions experiencing high strain, increasing the number of fixation elements and favouring running sutures over single sutures are critical adjustments. When abdominal wall loading varies significantly, a uniform fixation approach, such as “single crown”, may not be appropriate. Instead, localization should consider specific local displacements and distortions, particularly in areas with the highest strain. Recognizing the importance of energy dissipation, we frequently employ continuous running sutures as robust support along the steep tissue shifts highlighted in red in the leftmost images of Fig. 1. In summary, our reconstruction decisions are guided by the insights gained through HEDI’s imaging process. Mesh placement, type, size, fixation elements, and overlap are all derived from visualised data. We attribute the reduced pain levels and lower recurrence rate to the enhanced visualisation capabilities now available to surgeons.”

Indeed, we recently encountered a case involving a hernia patient who underwent treatment without considering HEDI's results. This patient developed a significant bruise at a corner of the mesh, resulting in substantial pain. HEDI's findings indicated that the mesh size used was insufficient to adequately cover the entire unstable abdominal wall, and the bruise occurred precisely within the highlighted unstable region.

Reviewer Comment C2.2. Discussion pretty short may want to add some more on this cool technology

Response. Thank you for your valuable input. Recognizing the potential for an extended discussion, we have taken your suggestions into account. The discussion has been substantially expanded, incorporating insights from your Comment C2.1, which addresses the influence on decision-making. Additionally, we've integrated perspectives from comments C1.1 (regarding the 15 mm threshold), C1.6 (establishment of a proper control group), and C1.8 (regarding robustness of our method) provided by reviewer one.

Response to Referees

COMMSMED-23-0436

HEDI: First-Time Clinical Application and Results of a Biomechanical Evaluation and Visualisation Tool for Incisional Hernia Repair

Reviewers' comments:

Reviewer #1 (Remarks to the Author):

Reviewer Summary. The work investigates the value of integrating biomechanical and geometrical data (by post processing CT images) for Incisional Hernia Repair. The authors present a method (with adequate literature references) and test its clinical value in a 62 ± 13 years old patient cohort, out of which 31 patients (age not reported) underwent surgery. All 31 patients remained pain-free, and showed no evidence of hernia recurrence after three years of follow-up.

Reviewer Comment C1.1.1. The definition of the “unstable area” by an absolute displacement of more than 15mm, needs further justification. Human anatomy varies largely, and at least a relative measure, i.e. relative to body size, or relative the mean abdominal wall displacement seems to be a better choice.

Response. Thank you for raising this crucial point. Our choice of a fixed 15 mm displacement threshold was initially based on literature data (Fig. 9 in Reference 19: Kallinowski et al. 2021), which demonstrated its efficacy in distinguishing between healthy and herniated or IPOM-repaired abdominal walls. Moreover, empirical evidence indicates that the mesh configuration becomes irreversibly altered when displacement exceeds approximately 15 mm. Additionally, we opted for an absolute value due to its simplicity and ease of comprehension. However, we acknowledge the potential for variability in this threshold among patients due to differences in abdominal wall characteristics, including tissue texture and elasticity. Consequently, we strongly encourage further investigations in this area. To account for such variability, HEDI offers flexibility by allowing users to set their own threshold based on their understanding of the individual patient’s anatomy when starting the program, as now mentioned in the discussion section.

Reviewer Comment C1.1.2. As this is a severe limitation, it should be part of study limitations’ discussion.

Response. This limitation has already been incorporated into the discussion section on study limitations (lines 193–202).

Reviewer Comment C1.2.1. As a displacement always refers to two configurations, it is unclear how the abdominal wall displacement referring to “rest” or during the “Valsalva maneuver” was computed. As reported in the manuscript, “the mask at rest is transformed into the Valsalva mask by optimising an error measure using a symmetric diffeomorphic registration method. The registration produced a displacement field consisting of vectors pointing from a source pixel to its destination.” That’s perfectly understandable, but it is one single displacement (field) that can be calculated. Hence, the question: what is the rest-displacement and what is the Valsalva maneuver-displacement? In Fig.4 it seems that the same displacement is plotted twice, and then labeled differently?

Response. Thank you for highlighting this section, which could potentially lead to misunderstandings. Technically, the registration process yields a forward projection (from rest to Valsalva state) and a backward projection (from Valsalva to rest state). As a result, we distinguish between the configuration at rest and the configuration during the Valsalva maneuver. To prevent any misconceptions, we have modified the Method section, explicitly stating that two separate vector fields exist and describing the two different projections as follows:

“For each 2D slice, the mask at rest is transformed into the Valsalva mask by optimising an error measure using a symmetric diffeomorphic registration method. This process generates a map that facilitates the transition from the state at rest to the Valsalva state (forward projection) and the reverse

transition from the Valsalva state back to the state at rest (backward projection).

The resulting vectors either point from a source pixel at rest to its corresponding location during the Valsalva maneuver or in the opposite direction. Vectors pointing away from the centre of the mask defined the outward displacement during the Valsalva maneuver, while inward displacements were set to zero, and vice versa for the backward projection.

Subsequently, both masks were then colour-coded based on the magnitude of the displacement vectors connecting each pair of points, resulting in two distinct 3D models showing the magnitude of translation between the two acquisition instances.”

Reviewer Comment C1.2.2. Given this explanation, it is obvious that both vector fields represent the same displacement information – either recorded outwards or inwards. It then explains why the two fields plotted in Figure 1 are so similar. The fact that both fields are not exactly the same is simply:

1. the consequence of not using identical pixels in the displacement computation, and
2. some data has been erased (outwards pointing vectors in the back projection, and inwards pointing vectors in the forward projection).

Consequently, the difference between both fields is purely “numerical” and has no physical cause whatsoever.

Response. It is true that both vector fields represent the same displacement information. We revised the manuscript to avoid confusion, clarifying that only one displacement field is determined and then mapped onto both surfaces at Rest and during Valsalva (see completely revised Method section “Symmetric diffeomorphic registration”, lines 267–292).

While the results look very similar, visualizing both is essential for complete information. For example, imagine a coin-sized area at rest that expands significantly during Valsalva (e.g., into a balloon-like object), while the surrounding area remains relatively unchanged. Visualizing only the rest state with a coin-sized area highlighted in red, yellow or white would not convey the full extent of this dynamic behaviour.

Additionally, we no longer erase vectors (e.g. inwards pointing vectors), as all movements (inwards and outwards) are important for understanding the behaviour.

Reviewer Comment C1.3.1. In relation to point 2: What is the strain associated with “rest” and “Valsalva maneuver”?

Response. The strain is calculated from the gradient of the corresponding displacement fields. As mentioned in our response to your Comment C1.2, we have one strain field for the configuration at rest and another for the Valsalva state. These are derived from the vectors of the forward and backward projections.

Reviewer Comment C1.3.2. No, there is no physical meaning in the difference between both displacement fields, and as such among their spatial gradients either. See also the response above.

Response. The strain tensor field is now calculated based on the “forward” displacement field and then mapped onto both the surface at Rest and the surface during Valsalva using the displacement vectors.

Reviewer Comment C1.4.1. Strain is calculated from the gradient of the displacement field and has clear physical meaning; normal strain reflects the change in length, and shear strain reflects the change of shape. As a matter of fact, a three-dimensional displacement field results in a six-dimensional strain field, which then prompts the question: which strain component, or combination of strain components, is shown in the figure/analyzed in this work. What does a strain level of 0.2 mean, for example? In addition, in relation to the selected frame of reference, many different strain definitions are known (engineering strain, Green-Lagrange-strain, Euler- Almansi strain, logarithmic strain,...). Which strain measure has been used in this work? This are not just semantic questions, but qualifies/disqualifies the applied method.

Response. That's an excellent observation. The local strain is defined by nine components, six of which are independent. We calculated the Euler-Almansi strain Tensor based on the displacement

field, following the methodology outlined in reference 18 (Abd-Elmoniem et al. 2008). This tensor is often reduced to a single local value to facilitate comparison with results from mechanical testing or for visualization purposes. In our study, we adopted the von Mises equation to compute a scalar field representing the strain. While this equation is typically associated with stress calculations, in this instance, we adapted it for strain analysis. For clarity and completeness, we have incorporated the following sentence into the Results section (lines 111-112):

“The local strain tensor was reduced to a single value equivalent to the von Mises equation for mechanical stress.”

Reviewer Comment C1.4.2.1 Ref 18 indeed worked with the Euler-Almansi strain, but interestingly called it 3-D Eulerian strain. However, it is essential that it is only the Euler-Almansi strain, given the displacement field stems from the forward projection. One CANNOT take the displacement field from the backward projection and put it into eq(4) of ref 18, i.e. the definition of the Euler-Almansi strain. The result would simply make no physical sense.

Response. That is true, as mentioned in the response before, we now use the forward displacement field to calculate the strain values and map them onto both surfaces.

Reviewer Comment C1.4.2.2. In addition, the computation of the average strain (application of the von Mises stress to the strain tensor) lacks physical meaning in the present context. One can easily show that a large strain value could in fact result from zero tangential strain in the skin. Instead using the maximum principal strain at the surface (in the the skin) would be a sensitive strain measure in the present context.

Response. We opted for maximum principal strain of the Green-Lagrange strain tensor and updated the text (lines 115–119) and Fig. 1 accordingly.

Reviewer Comment C1.4.2.3 In addition, the way the displacement fields have been generated, i.e. morphing 2d slices, neglects any strain that would appear in the third (out-of-plane) direction.

Response. We opted for calculation in 3D (see completely revised Method section “Symmetric diffeomorphic registration”, lines 267–292), which led to larger areas of the unstable abdominal wall (line 135).

Reviewer Comment C1.4.2.4 In addition, the displacement vector that is used in strain definitions has to connect the reference and spatial configurations of a single material particle. As morphing cannot guarantee that the two connected pixels belong to the same material particle, the resulting displacement is questionable in the computation of a strain field.

As a consequence, and in addition to the statement made in response to C1.2, the present methodology is severely flawed, and it does not make any sense to discuss results from such an analysis.

Response. While it is true that we cannot guarantee that two connected voxels belong to the same material particle, we have shown in our newly included evaluation (see response to Comment C1.8.2 and new Method section “Evaluation of Registration”, lines 294–304) that our approach provides a fast and reasonably accurate approximation of the actual physical behaviour during the Valsalva maneuver, and thus offers (for the first time) invaluable additional information for surgeons.

Reviewer Comment C1.5.1. The statement “Although HEDI offers a first step in visualising the stress-strain relationship of abdominal wall structures” in the discussion section, is unclear. The method, as detailed in the manuscript measures some sort of kinematic quantity (denoted by strain but unlikely to be any physically meaningful strain), but it does, for sure, not consider stress by any means. Stress is a measure of tissue loading, not deformation, and the outlined method is not capable of computing stress.

Response. We fully agree with this comment. This statement is not justified based on our approach. The connection between strain and stress has not been investigated or visualized. We have revised the statement as follows:

“Although HEDI offers a first step in visualising the local displacement and strain of abdominal wall structures, [...]”

Reviewer Comment C1.6.1. As indicated by the authors, pain and recurrences plague appears in a number of patients after repair of an abdominal defect. However, it remain still unclear what the observation (all 31 patients operated on in this study remained pain-free, and showed no evidence of hernia recurrence after three years of follow-up) means. Any comparison with a proper control group would have helped to answer the question of the clinical benefit.

Response. We appreciate your comment and acknowledge the importance of a robust control group to establish the superiority of the HEDI-based approach. The primary objective of this study is to introduce the HEDI tool, developed as part of our new approach, and demonstrate its feasibility in a cohort of 141 patients, with follow-up data available for 31 patients.

Notably, while our primary focus is on large hernias, our smaller patient subgroup with available follow-up data outperforms reported literature statistics, including chronic pain and high recurrence rates (reported as 22-32%) across all hernia types, regardless of size. Therefore, we are confident that our approach will potentially yield positive results when applied to a larger cohort.

However, to further solidify our findings and establish a proper control group, we plan to employ propensity score matching (PSM) with the HERNIAMED[®] registry once we gather a larger cohort, as now mentioned in the discussion section.

Reviewer Comment C1.6.2. As this is a severe limitation, it should be part of study limitations' discussion.

Response. This limitation has already been incorporated into the discussion section on study limitations (lines 203–208).

Reviewer Comment C1.7.1. An elaboration on the implications of the relatively low DICE coefficient reported in section “automatic segmentation” would have been desired.

Response. Thank you for this important comment. The lower accuracy observed in hernia volume segmentation results from the substantial variability in hernia characteristics, including differences in size, location, and internal structure, as well as the inherent challenge of accurately delineating it from the surrounding abdominal region. It's important to note that we identified a technical issue in the TensorFlow implementation for Multi-GPU, specifically related to Batch Normalization. In this issue, Batch Normalization was being performed on individual subbatches on each GPU rather than on the entire batch. To address this problem, we decided to use synchronized Batch Normalization layers (improvement from 0.45 to 0.49 for hernia volume). Furthermore, we found that data augmentation, specifically horizontally flipping and randomly rotating the images, had a positive impact on improving the Dice score for hernia segmentation (improvement from 0.49 to 0.58 for hernia volume).

However, it's essential to acknowledge that the Dice coefficient for hernia volume segmentation remains relatively low. As a result, we recommend that users of HEDI exercise caution and visually assess the segmentation performance of the hernia volume. This visual validation step should be conducted before relying on the measured statistics of the hernia volume for any post-processing or decision-making purposes, as now mentioned in this section.

Reviewer Comment C1.7.2. It would still be worth quantifying its meaning in term of error in volume measurements.

Response. Volume errors are now included in the revised segmentation section (lines 246–247).

Reviewer Comment C1.8.1. Statements/data concerning method validation would have been desired: the information shown in the color-coded images, how robust is it?

Response. Thank you for your valuable comment. First, HEDI's application demonstrated feasibility and calculation reproducibility when applied to the CT data of our 141 patients, except for 27 special cases detailed in the 'Inapplicable CT Scans' section of the Methods.

Second, during the development of HEDI, we compared different registration algorithms, with a specific focus on those capable of accommodating significant deformations and proven success in medical applications. Two methods were evaluated: one utilising B-splines (Reference 24: Voß et al.

2020) and the other employing symmetric diffeomorphic registration (Reference 23). Both approaches yielded similar and comparable results in our dataset. This consistency provided us with confidence in the algorithm's validity. The ultimate decision to use symmetric diffeomorphic registration from the DIPY package was based on both performance and licensing considerations.

However, assessing intra-patient variability remains challenging due to the limited frequency of patient scans, necessitated by the need to minimise radiation exposure. The robustness of unstable abdominal wall detection under repetition requires further investigation, ideally utilising non-invasive techniques such as MRI or laser scanning, as now mentioned in the discussion.

Reviewer Comment C1.8.2. Clearly, validation in patients (except cancer patients) is difficult. However, the system could be easily validated using phantoms, an exercise that would then provide valuable data to judge its robustness.

Response. We have evaluated the registration process by comparing manually measured displacements of landmarks (electrodes) placed on the abdominal surface in a regular grid with 5 cm spacing with the HEDI results for three patients with small, medium, and large displacements (see new Method section "Evaluation of Registration", lines 294–304, and newly added Figure 6). The mean absolute errors were 1.6 ± 1.6 mm, and the normalized errors averaged $4.6 \pm 4.3\%$ relative to the maximum displacement of the electrodes. Considering an average pixel size of 0.81×0.81 mm² and a slice thickness of 1 or 2 mm, these errors are within the expected range of human error for landmark placement.

Response to referees

To facilitate review, we provide line numbers that correspond to the revised manuscript with tracked changes. Please note that we have added additional references at the end of the response letter [R1–27], which extend beyond those cited in the main manuscript [M1–28].

Reviewer #1

R1C1: Given one displacement field one can only compute one strain field, and it is completely unclear why two strain fields are shown in Fig 1.

We refer to our previous responses to Comments C1.2.2 and C1.3.2. It is correct that only a single strain field is computed from the displacement field. However, for visualisation purposes, we map this same strain field onto both the surface at rest and the surface during Valsalva.

While strain physically occurs during the Valsalva maneuver, surgeons typically assess the abdominal wall at rest, both during surgery and in preoperative planning. By projecting the strain field onto the resting surface, HEDI highlights where deformation will occur under load, thereby making the biomechanical information more intuitive and clinically useful.

R1C2: For some reason the authors decided to switch to the Green-Lagrange strain. However, the response to referee text always talks about the Euler-Almansi strain. What strain measure has been used in the plots?

We apologise for the confusion. While we initially used Euler-Almansi strain in earlier stages of development, we later switched to Green-Lagrange strain, as noted in our response to Comment C1.4.2.2.

Any remaining references to Euler-Almansi strain in the response text were either part of the referee's comments or carried over from earlier review rounds. All plots and results shown in the current manuscript use Green-Lagrange strain. Both strain measures produced visually similar outputs in our application. We apologise for any inconsistency in the terminology.

R1C3: How did the strain field influence the treatment decision?

AQR1C3: The strain field guides the surgeon in the placement of suture lines acting as quilting seams. A quilting seam is a straight seam that connects the tissue of the transversal fascia and the mesh together. Both the fascia and the hernia mesh are stretchable material although with different Young modulus. To account for the unpredictable energy transfer between the two components, a stretchable yarn is used for the suture line, e.g. Monomax® made from 4-hydroxy-butyrate. This suture line has a stretchability of 50 %. In addition, the quilting seam is placed loosely in 10 – 20 loops aiming at a narrow zigzag stitch. An example is given in Fig. R1.

Strain assessment during tissue deformation is crucial to evaluate relationships between mechanical loading and functional changes in biological tissues [R1] Many biological tissues have a complex structure allowing them to function under demanding cyclic loading conditions. It is necessary to gain insight into tissue structure to understand tissue function and the interaction with hernia mesh. Ideally intact native tissues should be imaged in 3D and under physiological loading conditions [R2].

Our approach uses the strain provided by the patient and depicts changes upon loading in 2D and 3D.

It is important to note that the primary aim of this paper is to describe HEDI as a tool to support incisional hernia repair by visualising biomechanical strain and displacement. The decision-making process for each individual patient remains entirely in the hands of the treating surgeon. We have now stated at several occasions in the paper that HEDI is not serving as a deterministic decision-maker and served as an adjunct rather than a directive system (lines 45 ff., 165 ff., 189 ff., & 263 ff.). The decision how to treat a patient remains fully at the discretion of the operating surgeon and his expertise (lines 187-189, 198-199).

Fig. R1 Images taken 2022 of a 60 yr old patient, suffering from a short bowel syndrome and an incisional hernia after multiple incisions and re-operations after a failed cholecystectomy in 2002 (patient gave written consent to present the images)

Far left: frontal view. **2nd from left:** side view standing. **Middle:** 3D fusion image of supporting muscles (blue), hernia during Valsalva (red), abdominal area with a shift above 15 mm (red rimmed yellow area). **2nd from right:** strain field during Valsalva. **Right:** intraoperative photo with placement of suture lines.

R1C4-1: What data provided by HEDI has been used in the treatment decision?

AQR1C4-1: HEDI provides both 2D and 3D visualisations along with pointwise strain distributions of key abdominal wall structures under load. It quantifies areas with displacement exceeding 15 mm during Valsalva, highlighting zones of instability that inform mesh sizing, positioning, and fixation. Prior to HEDI, these assessments were performed manually, requiring over 12 repetitions and several hours to achieve acceptable accuracy (<5% error). HEDI significantly accelerates this process while providing a more comprehensive and reproducible biomechanical analysis to support treatment decisions.

R1C4-2: Why should that be superior to conventional decision making?

AQR1C4-2: Enhanced understanding of strain-bearing structures has consistently advanced surgical imaging and decision-making. CT data alone is insufficient unless key biomechanical insights are extracted in a standardised way. HEDI supports this by visualising tissue displacement and strain under physiological load, providing objective, reproducible metrics that inform mesh placement and fixation strategies.

Training surgeons in biomechanical reconstruction has already been shown to reduce pain and recurrence compared to conventional approaches in certified hernia centres [M28]. HEDI builds on this by offering precise visualisation of instability and strain zones, further improving preoperative planning. In our cohort of over 100 patients treated with biomechanical planning, including HEDI in later cases, we observed no chronic pain, rapid return to activity, and recurrence rates below 1% at three-year follow-up in a high-risk population [M25].

R1C4-3: At the moment, no evidence justifies the first sentence in the discussion section.

AQR1C4-3: The first sentence of the discussion states: “The study highlights the effectiveness of using HEDI to enhance preoperative evaluations for incisional hernia repair, providing a more comprehensive understanding of the biomechanical support required for a stable repair.”

This is grounded in prior work demonstrating that biomechanical planning, specifically the mesh-defect area ratio (MDAR), is a better predictor of repair success than hernia size alone [R3]. While manual MDAR assessment is possible, it is time-consuming and prone to variability. HEDI was developed to automate this process through CT imaging and strain analysis, as outlined in our earlier methodological publication [R4].

Since then, more than 100 patients were reconstructed taking results of HEDI into consideration [M25]. In the results section of Nessel et al. (2024), para 6, it is stated: „HEDI was not necessary for less complex repairs. The HEDI output is related to abdominal wall instability. The distortion field is calculated using a symmetric diffeomorphic registration method [...]. It was first applied in 2% of cases with a complexity score of 2, 6% in group 3, and 9% in group 4. In the most complex cases, one-third

of cases were assessed using HEDI. However, since HEDI became available in 2020, the last year of recruitment for this report, this does not reflect the true need. Today, every complex case is evaluated with HEDI before elective repair. This is done to gain insight into biomechanical parameters“.

In a cohort of patients with an expected recurrence rate of at least 60 %, we see less than 1 % recurrences after 3 years. Our surgeons know of no other tool to use for the preoperative evaluation of biomechanical parameters of the herniated abdominal wall presenting with multiple failed attempts and recurrences. Some of the patients have had five and more operations before being evaluated with the help of HEDI, being restored to normal life with no more recurrence so far.

We believe that the first sentence of the Discussion section is supported by evidence.

Nevertheless, we have moderated this sentence and highlight that the primary aim of this paper is to describe HEDI as a tool to support incisional hernia repair by visualising biomechanical strain and displacement. While we provide guidance on how HEDI results can be interpreted, the decision-making process for each individual patient remains entirely in the hands of the treating surgeon:

Lines 158 ff.: “The study ~~highlights the effectiveness of using~~ demonstrates how HEDI ~~to~~can enhance preoperative evaluations ~~s for~~in incisional hernia repair; ~~by~~ providing a more ~~comprehensive~~ detailed understanding of the biomechanical support required ~~for~~to achieve a stable repair. Importantly, the clinical application of HEDI took place in parallel with its development, meaning that the visual outputs and processing steps of the tool evolved over time. As such, the results presented here reflect a dynamic integration of HEDI into surgical planning, and not a standardised, retrospective application of a finalised tool. Rather than serving as a deterministic decision-maker, HEDI provided surgeons with an additional source of biomechanical insight that complemented, but did not replace, their clinical expertise and judgement.”

R1C5: Besides using HEDI, the manuscript reports the use of the GRIP concept. What is it and how was it used?

AQR1C5: GRIP stands for *Gained Resistance against Impact related to Pressure* and was first introduced in reference [R4]. It is a biomechanical concept used to quantify whether a hernia repair can withstand the cyclic pressure loads experienced in daily life, especially from actions like coughing, sneezing, or lifting.

Beginning in 2012, we conducted cyclic loading tests on various mesh materials based on intra-abdominal pressure peaks of up to 280 mmHg (~37.3 kPa), which can last over a second, using a self-built test bench [M12]. Postoperative monitoring showed some patients coughed over 400 times in the first 24 hours, justifying this load case. From these tests, we defined CRIP (*Critical Resistance to Impact Pressure*), the minimum mechanical resistance required for a stable repair. CRIP and GRIP critically depend on tissue distension and distortion related to the stress-strain-relation of the individual’s tissues at a given pressure level.

In clinical practice, we use HEDI to assess strain and distension in individual patients, especially in complex cases. These measurements are then integrated into the GRIP framework to evaluate whether the planned reconstruction provides sufficient mechanical stability (i.e., GRIP > CRIP).

This decision process is formalised in treatment algorithms [M12], [M25]. It is guided by three clinical questions:

1. Should the GRIP concept be applied?
2. Can instability and elasticity be assessed clinically, if not, use HEDI.
3. Does the planned repair provide GRIP > CRIP, if not, revise the strategy.

Only once GRIP exceeds CRIP is the reconstruction considered biomechanically sufficient for implementation.

R1C6: The flow chart in Figure 4 misses the strain data – why?

We omitted the strain data from Fig. 4 to maintain clarity and avoid visual overload. The figure is meant to provide a high-level overview of the HEDI workflow, not a detailed instruction or analysis. Since the strain field is derived directly from the displacement field shown, including it might have

added redundancy. We aimed to keep the figure simple and focused on illustrating the sequence of processing steps.

R1C7-1: Concerning the newly added evaluation of the registration process: What was the ground truth that allowed the authors to come up with a 1.6mm error in Figure 6?

As described in the section “Evaluation of Registration”, surface electrodes were manually identified in the image data and served as anatomical landmarks. The distances between corresponding electrodes in the rest and Valsalva scans were measured and compared to the displacement values computed by HEDI. The reported average registration error of 1.6 mm reflects the mean difference between these independently measured and computed displacements.

R1C7-2: In addition, placing electrodes defines a fundamentally different (more robust) registration problem.

We agree with the reviewer’s observation and have added a corresponding note to the “Evaluation of Registration” section. We now explicitly state that this approach may improve accuracy and encourage radiologists to consider it, as it is relatively easy to implement in clinical imaging protocols. Lines 330-331: “It is important to note that using electrodes can make the registration process more robust and may therefore be considered in clinical practice.”

Reviewer #3

R3C1-1: Could you write the basis of the statement “Moreover, empirical evidence indicates that the mesh configuration becomes irreversibly altered when displacement exceeds approximately 15 mm.” ?

AQR3C1-1: As there are no systematic studies on healing soft tissue under stimulation, findings on bone healing are consulted [R5]. In fracture healing, the effects of mechanical stability and mechanical stimulation have been extensively studied in vitro and in vivo using a variety of animal models and stimulators. Perren’s interfragmentary strain concept has been used to describe primary and secondary fracture healing and the development of pseudarthrosis [R6]. The theory suggests that the strain that leads to the failure of fracture healing and the development of pseudarthrosis represents the upper limit of the load that can be tolerated by the regenerable tissue. In our opinion, this limit for soft tissue is approximately in the range of the above-mentioned displacements and strains.

The healing of bone fractures is regulated by mechanobiological influences [R7]. It is likely that the strain field in a bone (tissue) defect is crucial for load absorption in a fracture gap [R8].

From this perspective, the development of an incisional hernia and a recurrence represents a ‘pseudoscarosis’, i.e. a scar failure to heal, in analogy to pseudarthrosis. There are still many unresearched areas in this field. Biomechanical research in the field of orthopaedics and trauma surgery for the treatment of pseudarthrosis (started more than 50 years ago) currently varies fracture gap widths between 0 and 60 mm, uses natural, simulated and synthetic bones and employs at least 27 different standardisation routines [R9]. In soft tissue healing and incisional hernia repair, standardisation of cyclic load research has just begun, but it is already evident that the above-mentioned limits are most likely subject to a variety of influences [R10].

If a ventral or dorsal displacement of more than 15 mm is perceived when viewing the abdomen from the side during a movement, a relevant elastic deformation of the abdominal wall must be assumed. HEDI quantifies the area of the abdominal wall subjected to this displacement. If such a distension is found, hernia mesh must withstand a relevant elastic deformation of the abdominal wall. While further research is needed, this empirical limit has proven practical in identifying high-risk areas during surgical planning, an example is described in **AQR3C3**.

We have adjusted this paragraph (lines 215 ff.) to: “Moreover, empirical evidence indicates that the mesh configuration becomes irreversibly altered when displacement exceeds approximately 15 mm. Moreover, if such distension is present, the hernia mesh must be able to accommodate the resulting elastic deformation of the abdominal wall, and it is likely that its configuration becomes irreversibly

altered once a certain threshold is exceeded. The precise value of this threshold, however, requires further investigation.”

R3C1-2: Is this clinical experience supported by some imaging? What happens to the mesh?

AQR3C1-2: Prior studies have demonstrated that repeated mechanical loading can alter the shape and positioning of implanted meshes, affect their edges, and influence surrounding tissue behaviour. These effects have been shown in both experimental and clinical models involving animal and human abdominal walls [R11–R18].

Although we did not conduct a systematic imaging study on mesh deformation, clinical experience supports these findings: using CT abdomen with Valsalva gives insight into the deformation and movement of meshes upon repeated force. Below an example is depicted for illustration: a cut-out part of the CT abdomen under Valsalva two years after surgery is given (left) and the mesh highlighted with arrows (right). Yellow arrows point to the mesh laying still straight and red arrows point to mesh deformation (Fig. R2).

Fig. R2 Mesh movement and deformation over time of a patient complaining of pain after hernia reconstruction with a mesh.

Another illustrative example involves a 50-year-old male, five years post-liver transplant and two years after HEDI-supported reconstruction (Fig. R3). Preoperative HEDI revealed an unstable abdominal wall region (red-rimmed yellow area with >15 mm displacement). Two years later, the patient reported mild pain during strenuous exercise. Imaging showed a slight bulge (~12 mm) during Valsalva but no hernia recurrence. Here, ultrasound confirmed the mesh remained intact and properly positioned. Given this, we attribute the bulge and sensation to elastic deformation of the mesh, likely due to localised mechanical strain.

This case highlights how even when meshes remain in place, dynamic deformation may still occur under load. Tools like HEDI help anticipate such outcomes by identifying high-strain regions during planning, allowing for better-informed mesh placement and fixation.

Fig. R3 HEDI-based evaluation and follow-up in a 50-year-old male patient, five years after liver transplantation and two years post HEDI-guided hernia reconstruction (written consent obtained for image publication). **Left:** Preoperative HEDI result showing the unstable abdominal wall area with >15 mm displacement, highlighted in the red-rimmed yellow region. **Left middle:** Patient indicating the location of mild pain during strenuous activity (e.g., playing soccer). **Middle right:** HEDI image at rest. **Right:** HEDI image during Valsalva maneuver, revealing a bulge of approximately 12 mm perpendicular to the abdominal circumference. Ultrasound confirmed that the mesh remained intact

and properly positioned; the observed bulge is therefore attributed to elastic deformation of the mesh under load.

R3C2: Could you comment on the limit referring to the healthy abdominal wall behavior.

Jourdan et al. 2022 showed the displacement of abdominal wall of healthy abdominal wall during Valsalva maneuver.

Jourdan , et al. "Dynamic-MRI quantification of abdominal wall motion and deformation during breathing and muscular contraction." *Computer Methods and Programs in Biomedicine* 217 (2022): 106667

AQR3C2: We thank you for pointing out this valuable manuscript to us. Jourdan et al state in the Material and Methods section: Twenty healthy subjects (8 women) were included after they provided informed written consent. The exclusion criteria included a history of abdominal orincisional hernia. In the Abstract and in the Results section, they state: The largest displacement was observed for the medial part of RA (17.9 ± 8.0 mm) whereas the posterior part of LM underwent limited motion (2.8 ± 2.3 mm) [M27]. This finding is in keeping with the limit of 15 mm chosen by us as a cutoff between healthy and herniated.

Our own work indicates a marked difference between healthy and herniated abdominal walls with a good cutoff between 10 and 20 mm (Fig. R4 given below as retrieved from reference [M12]). Further research will refine and individualise the 15 mm chosen by our group for HEDI.

Fig. R4 Histograms of maximum distension in healthy abdominal walls, herniated abdominal walls, and walls after IPOM repair (top to bottom; from Fig. 9 in reference [M12]). The original caption referred to 'maximum strain distributions,' although the values shown correspond to measured maximum distension, with the assumption that greater distension reflects higher strain. In most healthy and IPOM-repaired abdominal walls, maximum distension was <1.5 cm, whereas in herniated abdominal walls it exceeded 1.5 cm.

We apologise, we already mentioned this in the manuscript (line 211), but the reference provided by us pointed to the wrong study. We included both references as follows (lines 211 ff.): "Our choice of a fixed 15 mm displacement threshold was initially based on literature data^{49,12}, which demonstrated its efficacy in distinguishing **between herniated** from healthy **and herniated** or IPOM-repaired abdominal walls. Jourdan et al. further reported a maximum displacement of 17.9 ± 8.0 mm in healthy abdominal walls²⁶, supporting our selection of 15 mm as a cutoff between healthy and herniated tissues."

R3C3: Could you please explain in more detail why exceeding given displacement value is treated as a reason to overlap given area?

AQR3C3: As discussed in our response to R3C1-1, the 15 mm displacement threshold is informed by biomechanical principles from fracture research, where excessive strain impedes tissue healing [R6]. While this threshold provides a practical reference, we fully recognise that it simplifies a complex biomechanical reality. For this reason, HEDI allows users to customise the displacement threshold in 1 mm increments to adapt to individual cases and clinical judgement.

To illustrate the potential clinical relevance of exceeding this threshold, we present a single case example (with written consent for educational use). A 70-year-old patient with a small hernia opening but a large area of abdominal wall instability (>15 mm displacement) declined our proposed biomechanically calculated reconstruction (BCR) and instead underwent implantation of a smaller mesh tailored only to the hernia defect (Fig. R5).

The HEDI result showed that regions with displacement exceeding both 15 mm and 30 mm extended beyond the mesh coverage. Although the hernia opening itself was well covered, the patient developed a large, persistent seroma after two weeks which required treatment for several months, unusual in BCR-treated patients, where seromas are rare and typically resolve quickly. We suspect that unaddressed mechanical strain in the uncovered area contributed to this complication.

Importantly, this is only an individual case and generalisation is difficult. However, it exemplifies our rationale for overlapping mesh beyond the defect to cover areas of instability identified through HEDI, helping to reduce mechanical complications and improve long-term repair stability.

Fig. R5 Images of a 70-year-old patient, with the implanted hernia mesh outlined in green and bordered in blue, and the displacement threshold increased from 15 mm (**top row**) to 30 mm (**bottom row**). Only resting-state images are shown for comparison. **Middle:** The mesh does not cover the area with >15 mm displacement (top), and the region with >30 mm displacement extends through the right edge of the mesh. **Left:** Strain concentrations are visible along the lower mesh edges, suggesting localised mechanical stress. **Right:** The hernia opening is fully covered by the mesh, but instability outside the covered area remains.

R3C4: I think that you should write in the text what is the reference state in the calculation of displacement and strain when you show each strain/displacement map. I suppose you change the reference when showing “rest” and non-rest maps. From the response to another reviewer, I understand the sores of difference. However, I think that all of this should be explained in the text to the readers.

AQR3C4: We thank the reviewer for this comment. Only one displacement field is calculated, namely from the rest state to the Valsalva maneuver, and the reference is therefore not changed. The resulting displacement magnitude is then mapped onto both surfaces for visualisation. For example, if two corresponding points, A on the rest surface and B on the Valsalva surface, are connected by a displacement of X mm, the value X is displayed at both A and B. This approach allows surgeons to appreciate the regions of displacement relative to the surface they most often observe during surgery (the resting state), while still reflecting the deformation that occurs under load.

We updated the following paragraph (lines 312-317): “A single three-dimensional displacement field was then calculated, transforming the rest mask into the Valsalva mask using symmetric diffeomorphic registration¹⁸. The resulting vectors indicate the movement from a source voxel at rest to its corresponding location during the Valsalva maneuver. Both masks were converted into surface meshes and colour-coded based on the magnitude of the displacement vectors connecting each pair of points so that each point reflects the same deformation relative to the rest state.”

Reviewer #4

R4C1: Introduction: Incisional hernia repair is often associated with chronic pain and high recurrence rates of 22-32%¹⁻³. [This is mostly due to an insufficient mechanical strength at the mesh-tissue interface.] Despite the availability of various mesh types⁴, surgical techniques⁵, and fixation methods, each with its own advantages and disadvantages^{1,6}, [the success of the repair largely depends on the size and location of the mesh used.] If the mesh is too small or [fixed under tension], it can lead to postoperative complications. Please could you give references to justify the bracketed parts? If there are assumptions, they should be reformulated as such.

AQR4C1: Thank you for highlighting this. We agree that more research is needed to justify an ultimate statement concerning the insufficiency of mechanical strength at the mesh-tissue interface. Still, the phenomena are also observable under clinical conditions: using CT abdomen with Valsalva gives insight into the deformation and movement of meshes upon repeated force (see Fig. R2 and our answer to R3C1-2).

We have edited these sentences as follows, added references supporting the statement that the success depends in the size and location of the mesh, and removed “fixed under tension”. Lines 53-57 now read: “This is mostly potentially due to an insufficient mechanical strength at the mesh-tissue interface. [...] the success of the repair largely depends on the size and location of the mesh used^{12,25}. If the mesh is too small ~~or fixed under tension~~, it can lead to postoperative complications.”

R4C2: Line 73: What exactly do you mean by “composite” material? Are you referring to the mesh and the surrounding soft tissue before or after mesh integration?

AQR4C2: We use the term “composite” to describe both the initial assembly, mesh and native tissue immediately after surgery, and the integrated construct that evolves over time. Initially, no cross-links exist between the mesh and the host tissue, and mechanical properties are largely determined by the individual components. As healing progresses, collagen cross-linking and tissue ingrowth lead to a mechanically and biologically integrated structure. This integration changes over time during the remodelling phase, and the composite’s strength and behaviour may adapt based on local mechanical stimuli such as cyclic loading.

Therefore, we use the term “composite material” to capture both the pre-integration and post-integration states of the mesh–tissue complex, recognising that its mechanical and biological properties evolve significantly over the patient’s lifetime.

We have updated lines 73 ff. for clarification as follows: “These forces transfer energy into the elastic-plastic ~~structure of the composite material~~ of the integrated mesh and surrounding soft tissue, which can ultimately ~~leading to~~ result in mesh failure.”

R4C3: Line 74: The criteria the authors mentioned are not intuitive. Are these criteria needed to prevent sliding or to provide overall a good stability of the repair? Why does a mesh need to be secured in a low displacement region? During a Rives technique for example, the mesh is secured in the rectus muscle and/or the posterior rectus sheath, which is likely the region of the AW with the highest mobility (in terms of antero-posterior displacement). Yet, this technique is probably the safest today.

AQR4C3: Since biomechanical data on soft tissue are rare, we resort to analyses of biomechanically based bone reconstructions performed since decades. These data indicate that low mechanical stresses and minimal gradient of displacement between the proximal and distal bony segments are good for long-term durability [R19]. This is why a mesh needs to be secured in a low displacement region.

We found during our bench test work that these criteria are needed to prevent sliding and to provide overall a good stability of a repair [M12].

In inguinal hernia, mesh non-fixation had a 40 % increase in recurrences (14/1770 vs 10/ 2026 pts) [R20]. In ventral and incisional hernia, very little evidence is available from 10 trials with serious flaws (787 pts, ventral and incisional not separated) [R21].

The mesh-defect area ratio (MDAR) became available in 2016 [R22]. Assessing laparoscopic repair, univariate analysis showed a statistically significant higher recurrence rate in incisional hernia, BMI \geq 35, defect width $>$ 4 cm, defect area $>$ 20 cm², mesh overlap $<$ 5 cm and ratio of mesh area to defect area (MDAR) \leq 12. Multivariate logistic analysis revealed that MDAR was the only independent predictive factor for recurrence ($p < 0.002$) [R23]. In a recent study from Poland, analysis confirmed that MDAR was the only independent parameter for recurrence [R24]. IPOM-plus was recently compared to eTEP evaluating 74 patients. MDAR was higher in the eTEP group (21 vs. 12) [R25]. After 24 months, three patients had recurrences (1/ 31 eTEP, 2/ 23 IPOM-plus).

We lack randomised studies with larger patient numbers and we have no comparison to RIVES technique. As it stands, the scale tips towards larger MDAR.

R4C4: Line 78: Although it makes totally sense to normalize the results with respect to quality of the tissue, higher displacement under pulse loading may come from a higher elasticity of the tissue but also due to the anatomy (muscle/sheath thickness) or the physiology (high IAP).

AQR4C4: We agree that higher displacement may also come from these factors and included them in the text (lines 221-224):

“However, we acknowledge ~~the potential for variability in this threshold~~ that the exact limit may vary among patients due to differences in abdominal wall characteristics, ~~including tissue texture and elasticity~~ (e.g., tissue texture and elasticity), anatomical factors (e.g., muscle or sheath thickness), and physiological conditions such as elevated intra-abdominal pressure (IAP).”

R4C5: Line 88: How do you justify the relevance of using this 15-mm criteria identified on ex vivo porcine ex situ AW sample for living human patients? Given the inter-individual variability regarding the anatomy, tissue elasticity, IAP etc, using the same displacement-based criteria for all patients raises some questions. Why not using a strain-based criteria that would provide some normalisation?

AQR4C5: We appreciate the reviewer’s important point. As discussed in AQR3C1–AQR3C3, the 15 mm displacement threshold is based on a combination of theoretical reasoning, analogies to biomechanical principles from fracture repair, and clinical experience. In practice, a visible ventral or dorsal displacement of $>$ 15 mm during movement suggests significant elastic deformation of the abdominal wall, which HEDI quantifies in both 2D and 3D.

It is still unclear at which distention or distortion the mesh starts to move and refuses to go back in its original position as illustrated in Fig. R2. HEDI offers an additional source of information, and offers the ability to highlight instabilities for different thresholds empowering future research.

We opted for a preferably simple and during clinical practice easily verifiable value. For a surgeon, it is at any time possible to check where large displacements occur under load and correlate that with the results observed using dynamic CT (Valsalva maneuver) and HEDI. In contrast, a pre-calculated strain cannot be verified at the patient's bed.

We fully acknowledge that this fixed threshold cannot capture the full range of inter-individual variability in anatomy, tissue elasticity, or intra-abdominal pressure (see also answer AQR3C3). That is why HEDI allows users to adjust the displacement threshold in full millimetre increments, making it adaptable to clinical judgement and specific patient characteristics as mentioned in the Discussion section. We demonstrated the additional insight in Fig. R5.

Additionally, we agree that a strain-based criterion could offer a more normalised and individualised assessment. While our current focus has been on displacement for its clinical intuitiveness and practical application, we recognise the potential of strain metrics and have added a corresponding paragraph to the Discussion section (lines 228-232):

“Additionally, a strain-based criterion could provide a more normalised and individualised assessment. However, our current focus has been on displacement because it is simple, clinically intuitive, and easily verifiable in practice. Surgeons can directly observe where large displacements occur under load and correlate these findings with dynamic CT (Valsalva maneuver) and HEDI output. By contrast, pre-calculated strain values cannot be readily verified at the patient's bedside.”

R4C6: Lines 137-140: Given the size of the meshes used, I am presuming a component separation technique was performed for some patients. Could you say more about this? Excepted the size of the mesh, was a CST performed for other reasons (e.g., too much tension in the posterior rectus sheath/loss of domain)?

AQR4C6: Posterior component separation is performed as an integral part of our surgical toolbox whenever needed. We did not measure tension in the posterior rectus sheath. Loss of domain is attributed by the peritoneal flap technique (PFH) [R26]. Incisional hernia repair with the PFH technique is associated with a low risk of short and long-term complications [R27]. We addressed this risk with a pre- and rehabilitation scheme detailed in Nessel et al. [M25]. In this manuscript, in the highest-complexity group, 94% of patients underwent posterior component separation (= transversus abdominis release, TAR) augmented with a DIS class A mesh with non-resorbable suture fixation. In the HEDI group evaluated here, only four patients did not receive a TAR (non-TAR mesh size average \pm standard deviation, median: 719 ± 342 , 730 cm^2 vs TAR: 1053 ± 354 , 1060 cm^2 ; $p = 0.01468$). Mostly, TAR is needed for mesh size due to larger unstable abdominal wall areas, but concomitant parastomal repair and bypass circulation for liver cirrhosis are reasons as well.

R4C7: Line 162: Are you sure these strain hotspots are actual hotspots and are not derived from “side effects” during the registration/differentiation process. Fig 1a) 1b) show for example spots at the interface ribs/rectus muscle. Do the hotspots reveal a high-strain region or from a region with a high gradient between a low-motion region (ribs) and a high-motion region (muscle)? The strain field derived from a spatial differentiation of the displacement field which amplifies the noise already present in the displacement field. In other words, although a validation was performed on a continuous field (Evaluation of Registration) what level of validation does the image processing have (segmentation, registration, differentiation)?

AR4C7: We thank the reviewer for this important and insightful comment. We validated the segmentation of anatomical structures as described in the section “Automatic Segmentation.” However, to minimise registration errors, the symmetric diffeomorphic registration is applied only to the abdominal wall surface, extracted via a threshold-based method. While this approach avoids error propagation from multi-structure segmentation, it does carry limitations: segmentation inaccuracies may still arise due to incorrect thresholding or suboptimal image acquisition, particularly in the cases described in “Inapplicable CT Scans” and illustrated in Fig. 5.

We deliberately emphasised these limitations in the results shown in Figs. 1a and 1b to caution against over-reliance on HEDI outputs. The tool is intended to support, not replace, surgical judgement. It must be interpreted carefully and always in conjunction with clinical expertise.

The registration algorithm itself uses a symmetric diffeomorphic model, which is well-established for producing smooth, topology-preserving displacement fields. Validation using surface electrodes yielded a mean registration error of 1.6 mm, which we consider acceptable for clinical use.

We have clarified these points at several occasions (lines 45 ff., 165 ff., 187 ff., 198-199, & 263 ff.), highlighting that while HEDI offers valuable biomechanical insights, its output must always be interpreted within the broader anatomical and clinical context.

R4C8: Fig 2 (right), the defect seems located in the most cranial region of the AW. Also, the unstable region detected by HEDI seems to show that the most cranial part of the defect should ne be covered. Yet it is known that an insufficient mesh overlap in that region due to an insufficient sub-xiphoid dissection may lead to recurrence as well. In that case, what does HEDI recommend? Please specify.

AQR4C8: Thank you for the observation. It is important to clarify that HEDI is a decision-support tool, not a prescriptive system. It provides detailed visualisations of displacement and strain to assist surgical planning, but the final strategy remains entirely at the surgeon's discretion. In this sense, HEDI does not recommend but visualise. Importantly, HEDI does not indicate areas that should be excluded from coverage, instead, it highlights critical regions of instability that must not be overlooked during mesh placement and fixation.

In the specific case shown in Fig. 2 (right), dissection went along the diaphragm to the right and left crus, using the left diaphragmatic vein as a guide when we approach the left liver vein. The tendinous centre of the diaphragm was also utilised for mesh placement.

R4C9: Line 128: It is said that the HEDI and the GRIP concepts were used for this study. Based on the literature, the GRIP concept assesses the quality of the repair based on the repair characteristics (mesh type, size, number of fixations etc). Therefore, the recommendations provided by the GRIP concept are not unique. As different repair techniques could be equivalent from a GRIP standpoint (e.g., large mesh without fixation vs. smaller mesh with many fixations points), could the author specify which strategy in this study was used?

AQR4C9: Thank you for the insightful comment. You are correct that the GRIP concept allows for different combinations of mesh type, size, and fixation strategy to achieve biomechanical sufficiency. When multiple GRIP-compliant options were available, the preferred strategy was the one that minimised operative time, provided it still achieved GRIP > CRIP.

R4C10: Although this study seems to provide good results in terms of recurrence rate, it also raises some questions about the potential overkill associated with this technique. Would the patient in Fig1a with a small defect have complications with a mesh in the retro rectus area providing a 5-cm overlap? Although CST offers advantages (e.g., decreased tension in the midline, use of large mesh), it is also associated with limitations (e.g., limited long-term impact, invasiveness) that should be explicated here.

AQR4C10: Thank you for raising this important point. As discussed in AQR3C3 and illustrated in Fig. R5, HEDI allows us to visualise tissue displacement at adjustable thresholds, which helps avoid over-treatment by tailoring the repair to the patient's specific biomechanical profile. For example, in cases like the patient in Fig. 1a, where the hernia defect appears small, HEDI can reveal underlying instability that would otherwise go unnoticed with conventional assessment.

This individualised approach ensures that mesh size, placement, and the need for component separation are determined not by defect size alone, but by functional tissue behaviour under load. Among the more than 240 reconstructions performed to date, this strategy has helped us avoid both under-treatment and unnecessary invasiveness.

We have added the following sentence to the Discussion section highlighting HEDI's role in preventing over-treatment through biomechanical patient-specific treatment (lines 267-269): “However, careful assessment of abdominal wall instability is essential to confirm clinical suspicion, particularly in small hernia defects, and to avoid both over- and under-treatment in critical cases.”

R4C11: The absence of a “negative control” arm should be mentioned as a limitation of this study and a methodology providing a clinical validation of this approach should be provided.

AQR4C10: We have added the following paragraph to the limitations (lines 240-242): “Another limitation of this study is the absence of a formal negative control arm. Clinical validation could be pursued by comparing outcomes before and after Biomechanically Calculated Reconstruction (BCR)²⁵ training, evaluating results with and without the use of HEDI.”

References:

- R1 Olchanyi, M. D. *et al.* Validation of markerless strain-field optical tracking approach for soft tissue mechanical assessment. *Journal of Biomechanics* **116**, 110196 (2021).
- R2 Disney, C. M., Lee, P. D., Hoyland, J. A., Sherratt, M. J. & Bay, B. K. A review of techniques for visualising soft tissue microstructure deformation and quantifying strain *Ex Vivo*. *Journal of Microscopy* **272**, 165–179 (2018).
- R3 Kallinowski, F. *et al.* Assessing the GRIP of Ventral Hernia Repair: How to Securely Fasten DIS Classified Meshes. *Front. Surg.* **4**, 78 (2018).
- R4 Kallinowski, F. *et al.* Biomechanics applied to incisional hernia repair – Considering the critical and the gained resistance towards impacts related to pressure. *Clinical Biomechanics* **82**, 105253 (2021).
- R5 Jagodzinski, M. & Krettek, C. Effect of mechanical stability on fracture healing — an update. *Injury* **38**, S3–S10 (2007).
- R6 Perren, S. M. Optimierung der Stabilität flexibler Osteosynthesen mit Hilfe der Dehnungstheorie. *Orthopäde* **39**, 132–138 (2010).
- R7 Mehl, J. *et al.* External Mechanical Stability Regulates Hematoma Vascularization in Bone Healing Rather than Endothelial YAP/TAZ Mechanotransduction. *Advanced Science* **11**, 2307050 (2024).
- R8 Karali, A. *et al.* Full-field strain of regenerated bone tissue in a femoral fracture model. *Journal of Microscopy* **285**, 156–166 (2022).
- R9 Zhang, S. *et al.* Experimental testing of fracture fixation plates: A review. *Proc Inst Mech Eng H* **236**, 1253–1272 (2022).
- R10 Inglis, B. *et al.* Biomechanical duality of fracture healing captured using virtual mechanical testing and validated in ovine bones. *Sci Rep* **12**, 2492 (2022).
- R11 Podwojewski, F. *et al.* Mechanical response of animal abdominal walls in vitro: Evaluation of the influence of a hernia defect and a repair with a mesh implanted intraperitoneally. *Journal of Biomechanics* **46**, 561–566 (2013).
- R12 Podwojewski, F. *et al.* Mechanical response of human abdominal walls ex vivo: Effect of an incisional hernia and a mesh repair. *Journal of the Mechanical Behavior of Biomedical Materials* **38**, 126–133 (2014).
- R13 Todros, S., Pavan, P. G., Pachera, P. & Natali, A. N. Synthetic surgical meshes used in abdominal wall surgery: Part II—Biomechanical aspects. *J Biomed Mater Res* **105**, 892–903 (2017).
- R14 Kahan, L. G. *et al.* Combined in vivo and ex vivo analysis of mesh mechanics in a porcine hernia model. *Surg Endosc* **32**, 820–830 (2018).
- R15 Tomaszewska, A., Lubowiecka, I. & Szymczak, C. Mechanics of mesh implanted into abdominal wall under repetitive load. Experimental and numerical study. *J Biomed Mater Res* **107**, 1400–1409 (2019).
- R16 Schmidt, A. & Taylor, D. Erosion of soft tissue by polypropylene mesh products. *Journal of the Mechanical Behavior of Biomedical Materials* **115**, 104281 (2021).
- R17 Mueller, K. M. A. *et al.* Mesh manipulation for local structural property tailoring of medical warp-knitted textiles. *Journal of the Mechanical Behavior of Biomedical Materials* **128**, 105117 (2022).
- R18 Campbell, J. M. *et al.* The classification of hiatal shapes and their use as a marker for complexity, operative interventions, and recurrence. *Journal of Gastrointestinal Surgery* **28**, 1578–1585 (2024).

- R19 Savoldelli, C., Ehrmann, E. & Tillier, Y. Biomechanical assessment of different fixation methods in mandibular high sagittal oblique osteotomy using a three-dimensional finite element analysis model. *Sci Rep* **11**, 8755 (2021).
- R20 Kobayashi, F., Watanabe, J., Koizumi, M. & Sata, N. Efficacy and safety of mesh non-fixation in patients undergoing laparo-endoscopic repair of groin hernia: a systematic review and meta-analysis. *Hernia* **27**, 1415–1427 (2023).
- R21 Mathes, T., Prediger, B., Walgenbach, M. & Siegel, R. Mesh fixation techniques in primary ventral or incisional hernia repair. *Cochrane Database of Systematic Reviews* **2021**, (2021).
- R22 Tulloh, B. & de Beaux, A. Defects and donuts: the importance of the mesh: defect area ratio. *Hernia* **20**, 893–895 (2016).
- R23 Hauters, P. *et al.* Assessment of predictive factors for recurrence in laparoscopic ventral hernia repair using a bridging technique. *Surg Endosc* **31**, 3656–3663 (2017).
- R24 Kozan, R., Anadol, A. Z. & Sare, M. A new criterion to predict recurrence after laparoscopic ventral hernia repair: mesh/defect area ratio. *Pol Przegl Chir* **93**, 40–46 (2021).
- R25 Taşdelen, H. A. Comparison of outcomes of the extended-view totally extraperitoneal rives-stoppa (eTEP-RS) and the intraperitoneal onlay mesh with defect closure (IPOM-plus) for W1-W2 midline incisional hernia repair—a single-center experience. *Surg Endosc* **37**, 3260–3271 (2023).
- R26 Malik, A., Macdonald, A. D. H., De Beaux, A. C. & Tulloh, B. R. The peritoneal flap hernioplasty for repair of large ventral and incisional hernias. *Hernia* **18**, 39–45 (2014).
- R27 Nielsen, K. A. *et al.* Evaluation of risk factors associated with the peritoneal flap hernioplasty for complex incisional hernia repair - a retrospective review of 327 cases. *Hernia* **28**, 2301–2309 (2024).

Response to referees

Reviewer #4 (Remarks to the Author):

Thank you for your answers. One suggestion:

R4C9: Line 128: It is said that the HEDI and the GRIP concepts were used for this study. Based on the literature, the GRIP concept assesses the quality of the repair based on the repair characteristics (mesh type, size, number of fixations etc). Therefore, the recommendations provided by the GRIP concept are not unique. As different repair techniques could be equivalent from a GRIP standpoint (e.g., large mesh without fixation vs. smaller mesh with many fixations points), could the author specify which strategy in this study was used?

AQR4C9: Thank you for the insightful comment. You are correct that the GRIP concept allows for different combinations of mesh type, size, and fixation strategy to achieve biomechanical sufficiency. When multiple GRIP-compliant options were available, the preferred strategy was the one that minimised operative time, provided it still achieved GRIP > CRIP.

Please specify this in the manuscript as it is an critical factor in my opinion.

Response: We added the following paragraph to the “Clinical application” section (lines 260–265 in the tracked manuscript): “The GRIP concept assesses repair quality based on characteristics such as mesh type, size, and fixation points. Because different combinations can achieve equivalent biomechanical sufficiency, GRIP recommendations are not unique. For example, a large mesh with few fixations may be considered equivalent to a smaller mesh with multiple fixation points. When several GRIP-compliant options were available, the preferred approach was the one that minimised operative time.”